# COMPASSNAV:STEERING FROM PATH IMITATION TO DECISION UNDERSTANDING IN NAVIGATION

**LinFeng Li**[1,2*]   **Jian Zhao**[2,3*]
**Yuan Xie**[1]    **Xin Tan**[1†]   **Xuelong Li**[2†]

[1]East China Normal University
[2]The Institute of Artificial Intelligence (TeleAI), China Telecom
[3]Northwestern Polytechnical University
`llfeng@stu.ecnu.edu.cn`, `{xtan, yxie}@cs.ecnu.edu.cn`
`zhaoj90@chinatelecom.cn`, `xuelong_li@ieee.org`

## ABSTRACT

The dominant paradigm for training Large Vision-Language Models (LVLMs) in navigation relies on imitating expert trajectories. This approach reduces the complex navigation task to a sequence-to-sequence replication of a single correct path, fundamentally limiting the agent's ability to explore and generalize. In this work, we argue for and introduce a new paradigm: a shift from Path Imitation to Decision Understanding. The goal of this paradigm is to build agents that do not just follow, but truly understand how to navigate. We materialize this through two core contributions: first, we introduce Compass-Data-22k, a novel 22k-trajectory dataset.Its Reinforcement Fine-Tuning (RFT) subset provides a panoramic view of the decision landscape by annotating all feasible actions with A* geodesic distances. Second, we design a novel gap-aware hybrid reward function that dynamically adapts its feedback to decision certainty, shifting between decisive signals for optimal actions and nuanced scores to encourage exploration. Integrated into an SFT-then-RFT recipe, our CompassNav agent is trained not to memorize static routes, but to develop an internal "compass" that constantly intuits the direction to the goal by evaluating the relative quality of all possible moves. This approach enables our 7B agent to set a new state-of-the-art on Goal navigation benchmarks, outperforming even larger proprietary models, and achieve robust real-world goal navigation on a physical robot. Project page: https://linengcs.github.io/CompassNav

## 1 INTRODUCTION

The goal of Embodied AI extends beyond building agents capable of robust, autonomous operation within complex, previously unseen environments; it aims to create collaborative partners capable of interpreting human intent and processing abstract natural language instructions. A key benchmark for this capability is Goal-Driven Navigation (Chaplot et al., 2020; Krantz et al., 2022). Unlike Vision-and-Language Navigation (VLN) (Anderson et al., 2018; Zhao et al., 2026), which provides dense, step-by-step instructions, goal-driven navigation presents agents with sparse, high-level goals (e.g., "find a chair"). This requires the agent to explore effectively and reason spatially under uncertainty, without explicit guidance.

Approaches to this problem generally fall into Modular or End-to-End (E2E) categories. Traditional modular systems rely on explicit world representations (Zhou et al., 2025; Yokoyama et al., 2024a). While geometrically precise, they face two main challenges in real-world deployment: fragility to sensor noise and a semantic gap where engineered maps struggle to interpret unstructured human

---

[*]These authors contributed equally to this work.
[†]Corresponding authors.

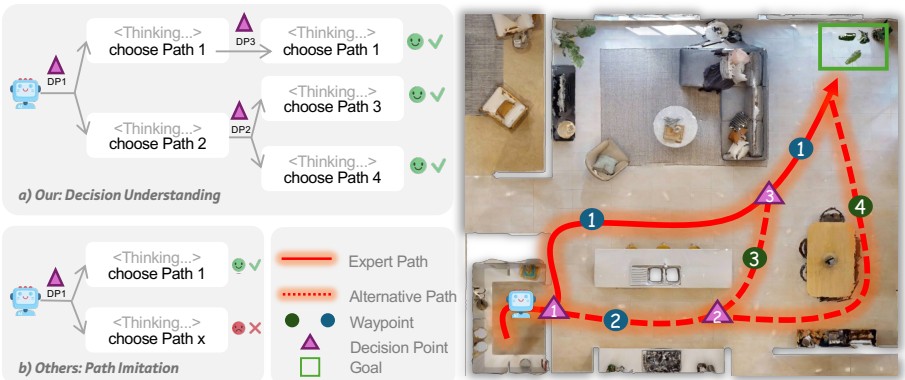

Figure 1: Visualization of the shift from Path Imitation to Decision Understanding. (Bottom Left) The prevailing Path Imitation paradigm treats navigation as replicating a single expert path (solid line), penalizing alternative choices. (Top Left) In contrast, our Decision Understanding paradigm teaches the agent to evaluate the relative value of all alternative paths (dashed lines), enabling flexible judgment at decision points. (Right) A concrete instantiation of these concepts in a navigation scenario.

commands (e.g., "find a quiet spot for reading"). In contrast, we advocate for E2E approaches (Goetting et al., 2024; Gu et al., 2025) based on Large Vision-Language Models (LVLMs). LVLMs avoid the reliance on fragile explicit mapping and possess a native capacity for processing natural language, making them well-suited for understanding complex human intent.

However, despite the potential of LVLM architectures, current training paradigms often rely on Path Imitation (Liu et al., 2025b; Gao et al., 2025; Krantz et al., 2020). This approach treats navigation as a memory task, optimizing the agent to minimize deviations from a single expert trajectory. We argue that this is suboptimal for dynamic environments where valid paths are rarely unique. Strictly imitating a single ground truth ignores alternative viable routes and fails to teach the agent the underlying causal structure of navigation—effectively prioritizing memorization over reasoning.

To address this, we introduce a paradigm shift towards Decision Understanding. We define this as the agent's ability to internalize spatial structures and execute a Think-then-Act heuristic. Unlike the binary signals used in path imitation, decision understanding requires the agent to analyze the current state, predict the outcomes of all available actions, and make decisions based on a panoramic value assessment. This enables the agent to learn not just what to do, but why one action is preferable to another. We visualize this shift in Figure 2.

We implement this vision in the CompassNav framework, which is built on two technical pillars. First, the Compass-Data-22k dataset moves beyond single-path supervision. Its RFT subset utilizes A* geodesic distances to annotate all feasible actions, creating a dense field of correctness within the decision space. Second, we design a Gap-Aware Hybrid Reward function to balance certainty and exploration. By analyzing the gap between optimal and suboptimal actions, the function enforces decisive strategies in clear scenarios while allowing for higher entropy in ambiguous ones, thereby adapting to the multi-path nature of real-world navigation.

Finally, to ensure our agent is prepared to develop this sophisticated reasoning capability, we address the "cold-start" problem (Zhang et al., 2025b) with a preparatory Supervised Fine-Tuning (SFT) stage. This entire SFT-then-RFT recipe provides an integrated solution that operationalizes our new paradigm, enabling agents to learn how to weigh options and decide, rather than just follow. Our core technical contributions are:

- **The Compass-Data-22k Dataset for Decision Understanding.** We introduce a novel dataset of 22k trajectories designed to support our new paradigm. It includes distilled reasoning traces for Supervised Fine-Tuning (SFT) and a Reinforcement Fine-Tuning (RFT) subset that shifts supervision from single-path imitation to a panoramic view, densely annotating all feasible actions with geodesic distances.

- **A Gap-Aware Hybrid Reward for Nuanced Policy Alignment.** We design a reward function that dynamically adapts feedback based on decision certainty. It handles ambiguity

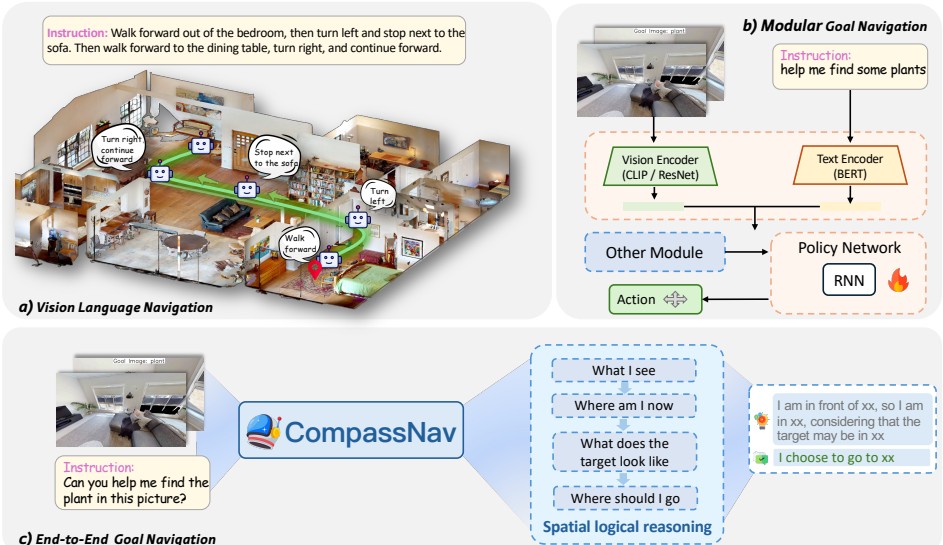

Figure 2: A conceptual comparison of embodied navigation paradigms.

using relative scores while providing decisive signals in clear-cut scenarios, effectively cultivating decision-making skills by leveraging the dense annotations in Compass-Data-22k.

## 2 RELATED WORK

**Modular vs. End-to-End Navigation.** Goal-driven navigation has traditionally been tackled by modular pipelines that strictly separate perception, mapping, and planning (Yokoyama et al., 2024a). Systems like CogNav (Cao et al., 2024) and UniGoal (Yin et al., 2025) rely on constructing explicit, engineered world representations—such as SLAM-based metric maps or topological graphs—to achieve high geometric precision. However, this reliance on explicit mapping introduces significant vulnerability in real-world deployment. Modular pipelines often suffer from error propagation, where sensor noise and pose estimation drift degrade the integrity of the constructed map, alongside the high engineering complexity required to integrate disparate modules.

Conversely, present End-to-End approaches utilizing LVLMs aim to map observations directly to actions. Early attempts leveraged powerful closed-source models (e.g., GPT-4o) for zero-shot navigation (Zhang et al., 2025a; Goetting et al., 2024). While demonstrating impressive reasoning, their prohibitive cost and latency render them impractical for real-time control. Alternatively, fine-tuning open-source LVLMs (e.g., Nav-R1 (Liu et al., 2025b)) has been hampered by a supervision granularity mismatch. Existing datasets (e.g., R2R) provide only single ground-truth paths, forcing models into a rigid path imitation paradigm that lacks the counterfactual data essential for robust decision-making. **CompassNav** overcomes these limitations by employing a novel training paradigm that distills robust spatial reasoning and decision-making capabilities directly into open-source LVLMs. This approach not only surpasses the navigation performance of expensive proprietary models but also contributes a cost-effective solution to the open-source community, significantly lowering the barriers to deploying advanced embodied agents.An intuitive comparison of these concepts is visualized in Figure 2.

**Reward Design for Reinforcement Fine-Tuning.** Effective RFT depends on reward quality. Sparse rewards (Yang et al., 2024) are intractable for long horizons, while simple dense heuristics (Luo et al., 2025) (e.g., Euclidean distance) fail to account for obstacles. Similarly, standard preference methods (Rafailov et al., 2024) rely on binary signals that miss the nuance of magnitude and ambiguity in decision-making. Consequently, current navigation RFT methods (Liu et al., 2025b; Qi et al., 2025) remain confined to a restrictive imitation paradigm, rewarding fidelity to a *single reference trajectory*. This approach reinforces rigid path replication rather than fostering the genuine decision understanding required for multi-path navigation.

## 3 THE COMPASS-DATA DATASET: GENERATION AND FORMULATION

We frame goal-driven navigation as a sequential decision-making problem. At each step, an agent observes its current view, is presented with a set of feasible candidate actions, and must select one to execute (further details are in Appendix B). This section details the generation and formulation of our novel dataset, Compass-Data, which is specifically designed to shift the training paradigm from path imitation to decision understanding, as illustrated in Figure 3.

### 3.1 COMPASS-DATA-RFT: DENSE ANNOTATION FOR DECISION UNDERSTANDING

Existing VLN datasets, such as R2R (Anderson et al., 2018) and RxR (Ku et al., 2020), are ill-suited for this purpose due to inherent structural limitations. Beyond the goal-suitability mismatch—where trajectory endpoints are defined by language constraints rather than object proximity—these datasets suffer from a critical supervision-granularity mismatch. By isolating a single ground-truth path without annotating alternatives, they constrain agents to rigid imitation. This lack of dense, comparative feedback prevents the model from evaluating the relative quality of the full action space, a necessary condition for learning genuine decision-making logic.

To facilitate "Decision Understanding," we constructed the Compass-Data-22k dataset, directly addressing the supervision scarcity inherent in existing benchmarks. Our generation pipeline transforms the training signal from a sparse optimal path into a panoramic value assessment (Figure 3). Specifically, the pipeline employs an Action Proposal Module (APM) (more details in Appendix C.1) that utilizes real-time depth maps and occupancy grids to identify all feasible candidate actions, discretizing them into polar vectors $(r, \theta)$. Distinct from traditional methods that only annotate the single best move, our Oracle $A^*$ Annotator leverages global simulator information to calculate shortest geodesic distances for every candidate action relative to the goal. This produces a complete action-value vector at each timestep, effectively mapping the gradient of correctness across the decision space. To further enhance data diversity, we incorporate a Backtracking Mechanism; rather than strictly adhering to the shortest path, the agent identifies Ambiguous Points—states with multiple viable options—and actively retraces its steps to explore and record alternative trajectories. This results in dense, panoramic supervision that teaches the agent the relative quality of all potential choices.

**RFT Data Structure.** After generation, this densely annotated data is structured for the Reward Fine-Tuning stage. To maintain a consistent structure throughout our framework, we establish a standard data format. As shown in Figure 3 (top right), each RFT data sample consists of a standard input (the instructional prompt and the agent's current visual observation) paired with a specialized target for reward modeling. This target contains the optimal action's ID and the complete vector of A* distances for all candidate actions at that step. This vector provides the fine-grained preference signal that is essential for our gap-aware hybrid reward function and the GRPO framework.

### 3.2 COMPASS-DATA-SFT: KNOWLEDGE DISTILLATION FOR POLICY INITIALIZATION

Initializing RFT directly from a base LVLM is often inefficient due to the "cold-start" problem (Zhang et al., 2025b). A base model's initial policy is typically poor, leading to suboptimal actions and sparse reward signals that hinder learning. To address this, we introduce a preparatory Supervised Fine-Tuning (SFT) stage using our Compass-Data-SFT-11k set, which is designed to instill a foundational "reason-then-act" capability.

Instead of constructing post-hoc reasoning for a predefined path, we adopt a more ecologically valid knowledge distillation strategy. We task a powerful teacher model, Qwen-QvQ (Team, 2024), to genuinely perform the ObjectNav task in `habitat-sim`. By recording the complete, step-by-step reasoning and action choices only from its successful episodes, we form an SFT dataset that reflects emergent and effective exploratory strategies.

**SFT Data Structure.** As shown in Figure 3 (bottom right), each SFT training instance shares the same input structure as the RFT data: (a) an instructional prompt and (b) the agent's current visual observation. Its distinction lies in the target output, which is a single string containing the teacher's full cognitive process and decision, formatted as `<think>...reasoning...</think><answer>k</answer>`. This format explicitly

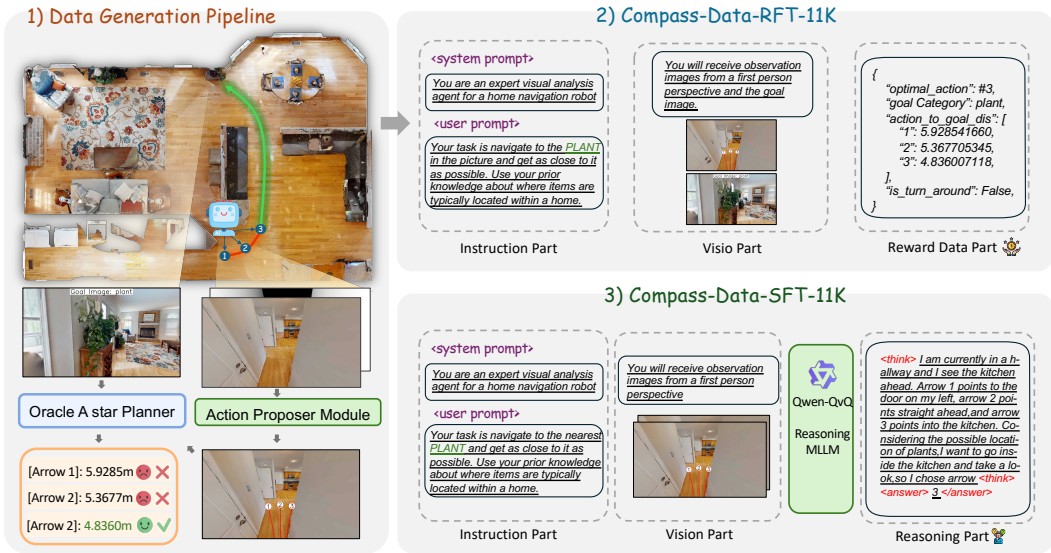

Figure 3: **Overview of our data generation and formulation pipeline.** (1) Data Generation: We use an A* planner in `habitat-sim` to densely annotate all feasible actions with their distance to the goal. (2) Dataset Formulation: Trajectories are formatted into two types. SFT data (bottom right) contains a teacher's reasoning trace for imitation. RFT data (top right) contains the full vector of action distances for reward modeling.

trains the model to externalize its reasoning before committing to an action, establishing the foundational 'reason-then-act' behavior. We intentionally exclude historical frames from the input to ensure compatibility with external memory modules. This design choice is deliberate, as our empirical validation showed that current methods for integrating historical context directly into the training process, such as using text summaries or concatenated images, consistently led to a degradation in performance (Details can be found in Appendix E.2).

## 4 COMPASSNAV FRAMEWORK

CompassNav is a two-stage fine-tuning recipe designed to first initialize a robust policy and then align it with environmental objectives. We adopt this SFT-then-RFT structure to create a synergistic learning process. The initial SFT stage efficiently overcomes the "cold-start" problem by instilling a strong policy prior through imitation. Subsequently, the RFT stage leverages this competent initial policy to align the agent towards genuine decision understanding. The entire process is visually summarized in Figure 4.

### 4.1 STAGE 1: POLICY INITIALIZATION VIA SUPERVISED FINE-TUNING

While our ultimate goal is to move beyond imitation, a foundational reasoning capability is a prerequisite. Stage 1 thus employs a targeted form of imitation (SFT) to instill this 'reason-then-act' structure by learning from a teacher model's reasoning process. At each timestep $t$, the model is trained to generate a two-part response that mimics the teacher's output: a chain-of-thought rationale followed by a final action choice, formatted as `<think> ... </think>    <answer>k </answer>`.

A key practical challenge is ensuring the model's action choice 'k' is always a valid action from the set of available candidates $\mathcal{A}_t$. To enforce this, we employ masked multiple-choice decoding. This is achieved by applying a masked softmax over the decoder's output logits $\mathbf{z}$ for the answer token, restricting the vocabulary to only the set of valid candidate indices $\mathcal{V}_t$:

$$\pi_\theta(j \mid x_t) = \frac{\exp(z_j)}{\sum_{j' \in \mathcal{V}_t} \exp(z_{j'})} \quad \text{for } j \in \mathcal{V}_t \tag{1}$$

This simple and effective technique ensures that all generated outputs are executable, which is crucial for the stability of the subsequent RFT stage.

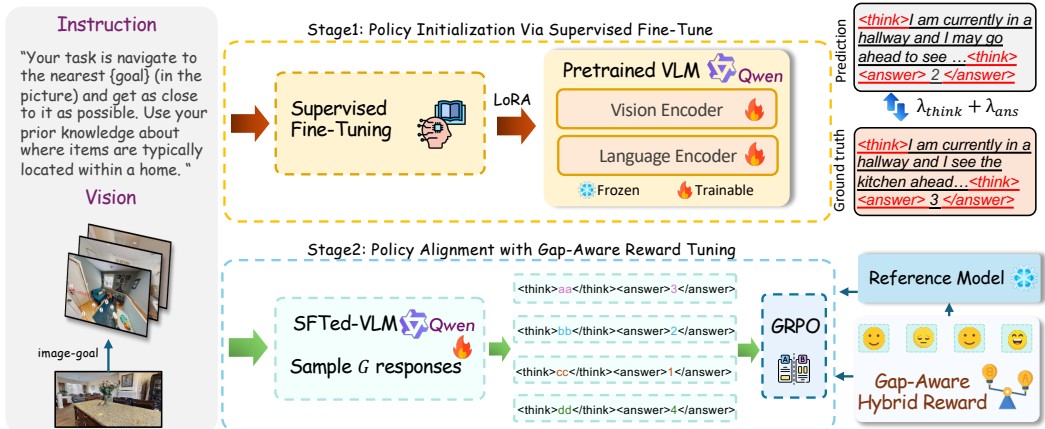

Figure 4: **The CompassNav two-stage training pipeline.** In Stage 1 (SFT), a pretrained VLM is fine-tuned to imitate a teacher's "reason-then-act" output. In Stage 2 (RFT), the SFT-tuned policy generates multiple responses, which are scored by our Gap-Aware Hybrid Reward function.

The SFT loss is a standard cross-entropy loss over the entire teacher-generated sequence, including both the reasoning tokens and the final action token:

$$\mathcal{L}_{\text{SFT}}(\theta) = \mathbb{E}_{(x_t, y_t)} \left[ \sum_{u=1}^{|y_t|} - \log p_\theta\big(y_{t,u} \mid x_t, y_{t,<u}\big) \right] \tag{2}$$

where $y_t$ represents the full target sequence `<think>...k</answer>`.

## 4.2 STAGE 2: POLICY ALIGNMENT WITH GAP-AWARE HYBRID REWARD TUNING

In Stage 2, we align the SFT-initialized policy with environmental objectives using the Group-wise Reward Policy Optimization (GRPO) (Shao et al., 2024) framework, as shown in Figure 4 (Stage 2). The learning signal is provided by our novel gap-aware hybrid reward function.

**GRPO Sampling and Reward Assignment.** For a given input prompt $x_t$, we use the policy $\pi_\theta$ to generate a group of $G$ distinct output sequences, $\{y_1, y_2, \ldots, y_G\}$. For each generated sequence $y_j$, we parse it to extract the chosen action $i_j$. The reward for this sequence, $r(y_j)$, is then determined by the quality of this chosen action, evaluated using our gap-aware hybrid reward function and the pre-computed A* distances for all candidates.

**Gap-Aware Hybrid Reward Function.** decisive signal in high-certainty scenarios while remaining nuanced to encourage exploration in low-certainty ones. This is achieved by composing the reward from two key components: a continuous base score and a certainty-modulated dynamic bonus.

The base score, $s_t^{(i_j)}$, provides a continuous evaluation for all available options, is calculated using a softmax function over the vector of distances to the goal, $d_t$. Actions with shorter distances receive a higher score, reflecting their relative quality:

$$s_t^{(i_j)} = \frac{\exp(-d_t^{(i_j)}/\tau)}{\sum_{k \in \mathcal{A}_t} \exp(-d_t^{(k)}/\tau)}, \tag{3}$$

where $\tau$ is a temperature hyperparameter that controls the sharpness of the distribution.

The core of our adaptive mechanism lies in the dynamic bonus, which is governed by a "certainty" factor, $g_t$. This factor dynamically assesses whether the current situation is decisive or ambiguous by measuring the normalized gap between the best ($d_t^{(1)}$) and second-best ($d_t^{(2)}$) options:

$$g_t = \text{clip}\left( \frac{d_t^{(2)} - d_t^{(1)}}{|d_t^{(1)}| + \epsilon}, 0, 1 \right). \tag{4}$$

This normalization allows the signal to prioritize broad exploration when far from the goal and promote precision when near it. A high $g_t$ indicates a clear, high-certainty choice, while a low $g_t$ signifies an ambiguous one.

The final hybrid reward, $r_t^{(i_j)}$, combines the base score with the bonus, which is modulated by this certainty factor and is triggered only for the optimal action $i^*$:

$$r(y_j) \equiv r_t^{(i_j)} = s_t^{(i_j)} + \beta_{\max} \cdot g_t \cdot \mathbb{K}[i_j = i^*], \tag{5}$$

where $i^* = \arg\min_k d_t^{(k)}$ is the optimal action, $\mathbb{K}[\cdot]$ is the indicator function, and $\beta_{\max}$ is the preset maximum bonus ratio. The final reward value is then clipped to the range $[0, 1]$.

To illustrate its superiority, we analyze its behavior against two common baselines—a sparse Binary reward (Rafailov et al., 2023) and a linear normalized Min-Max reward (Patro & Sahu, 2015)—across three key navigation scenarios in Figure 5.

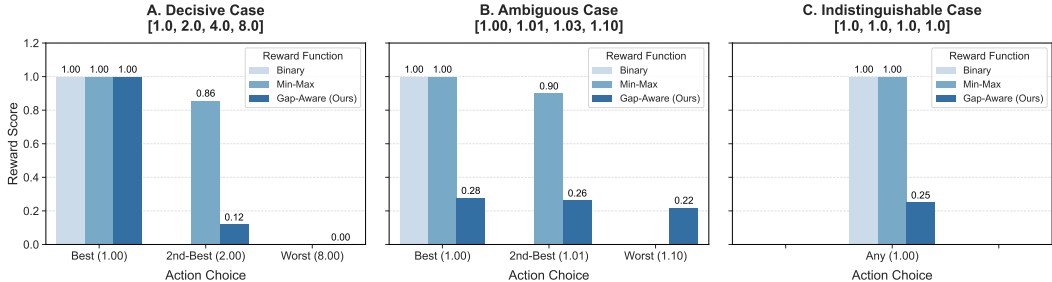

Figure 5: A comparative analysis of our Gap-Aware hybrid Reward against Binary and Min-Max schemes across three representative navigation scenarios. Each scenario shows the reward assigned for choosing the best, second-best, and worst actions.

As shown in Figure 5, our Gap-Aware hybrid reward function demonstrates superior behavior in diverse scenarios. In the Decisive Case (A), it creates a large margin between the best and second-best actions (1.00 vs. 0.12), providing a strong and unambiguous learning signal where baselines fail to differentiate. In the Ambiguous Case (B), it correctly assigns similar, non-extreme scores to closely-valued options, encouraging exploration rather than arbitrarily penalizing a viable choice. Finally, in the Indistinguishable Case (C), while other methods output a misleadingly perfect score of 1.0, our function provides an honest, low score of 0.25. This prevents the agent from receiving an illusory high reward for a random guess, fostering a more robust and faithful policy.

**Objective Function.** The GRPO objective maximizes the expected reward over the generated group. After normalizing the rewards to produce advantages $A(y_j)$, the loss function to be minimized is:

$$\mathcal{L}_{\text{GRPO}}(\theta) = -\mathbb{E}_{x_t, \{y_j\}} \left[ \sum_{j=1}^{G} A(y_j) \log \pi_\theta(y_j | x_t) \right] + \beta_{\text{KL}} \cdot \mathbb{E}_{x_t} \left[ \text{KL}(\pi_\theta(\cdot|x_t) \| \pi_{\text{SFT}}(\cdot|x_t)) \right]. \tag{6}$$

Here, $\pi_{\text{SFT}}$ is the frozen reference policy from Stage 1 (the "Reference Model" in Figure 4), and the KL term regularizes the policy update. This objective encourages the policy to generate sequences leading to high-reward actions, as evaluated by our fine-grained reward model.

## 5 EXPERIMENTS

### 5.1 EXPERIMENTAL SETUP

**Datasets and Tasks.** We generate training data in `habitat-sim` using the HM3Dv2 (Yadav et al., 2023) train split. To evaluate generalization, we test our agent on three challenging, unseen validation splits: HM3Dv1-val (Ramakrishnan et al., 2021a), HM3Dv2-val (Yadav et al., 2023) and MP3D-val (Chang et al., 2017). These splits feature completely held-out scenes and object instances, ensuring a rigorous evaluation of the agent's ability to navigate novel environments. The primary tasks are Object-Goal (Chaplot et al., 2020) and Instance-Image-Goal Navigation (Krantz et al., 2022; Liu et al., 2025a). Further details on dataset statistics and task definitions are provided in Appendix B.

**Evaluation Metrics.** We report two standard metrics: Success Rate (SR) and Success weighted by Path Length (SPL). SR measures the rate of successful episodes, while SPL weights each success by the ratio of the optimal path length to the actual path taken.

Table 1: Comparison with Modular goal-driven navigation methods. We categorize methods by whether they are End-to-End (E2E) and whether they utilize Memory (ME) (e.g., semantic maps, topological map).

| Method | Venue | E2E | ME | Backbone Models | | HM3D | | MP3D | |
| | | | | Vision | Text | SR | SPL | SR | SPL |
|---|---|---|---|---|---|---|---|---|---|
| Habitat-Web | CVPR'22 | ✗ | ✗ | - | - | 41.5 | 16.0 | 31.6 | 8.5 |
| ESC | ICML'23 | ✗ | ✓ | - | ⑤ GPT-3.5 | 39.2 | 22.3 | 28.7 | 14.2 |
| L3MVN | IROS'23 | ✗ | ✓ | - | ⑤ GPT-2 | 50.4 | 23.1 | 34.9 | 14.5 |
| InstructNav | CoRL'24 | ✗ | ✓ | ⑤ GPT-4V | LLaMA3-70B | 50.0 | 20.9 | - | - |
| PSL | ECCV'24 | ✗ | ✗ | CLIP | - | 42.4 | 19.2 | 18.9 | 6.4 |
| VoroNav | ICML'24 | ✗ | ✓ | BLIP | ⑤ GPT-3.5 | 42.0 | 26.0 | - | - |
| Pixel-Nav | ICRA'24 | ✗ | ✓ | LLaMA-Adapter | ⑤ GPT-4 | 37.9 | 20.5 | - | - |
| VLFM | ICRA'24 | ✗ | ✓ | BLIP-2 | - | 52.4 | **30.4** | 36.4 | 17.5 |
| GAMap | NeurIPS'24 | ✗ | ✓ | CLIP | ⑤ GPT-4 | 53.1 | 26.0 | - | - |
| SG-Nav | NeurIPS'24 | ✗ | ✓ | LLaVA-1.6-7B | ⑤ GPT-4 | 54.0 | 24.9 | 40.2 | 16.0 |
| UniGoal | CVPR'25 | ✗ | ✓ | LLaVA-1.6-7B | LLaMA-2-7B | 54.5 | 25.1 | 41.0 | 16.4 |
| CompassNav | ICLR'26 | ✓ | ✗ | ⑤ Qwen2.5-VL-7B | | **56.6** | 27.6 | **42.0** | **17.5** |

Table 2: Comparison of CompassNav with various open-source and proprietary models. Our 7B parameter model is benchmarked against models of varying scales.

| Models | ObjNav | | InsImageNav | | AVG | |
| | SR | SPL | SR | SPL | SR | SPL |
|---|---|---|---|---|---|---|
| *Open-source Models* | | | | | | |
| Qwen2-VL-7B (Wang et al., 2024a) | 35.9 | 14.8 | 5.30 | 3.60 | 20.6 | 9.20 |
| Qwen2.5-VL-3B (Bai et al., 2025) | 25.8 | 10.3 | 8.70 | 3.20 | 17.3 | 6.80 |
| LLama3.2-11B (Dubey et al., 2024) | 25.8 | 8.4 | 3.00 | 2.50 | 17.1 | 5.50 |
| *Closed-Source Models* | | | | | | |
| GPT-4o (Hurst et al., 2024) | 52.4 | 23.5 | 29.8 | 13.2 | 41.1 | 18.4 |
| Gemini-2.5-Flash (Comanici et al., 2025) | 50.0 | 10.6 | 24.1 | 9.70 | 37.1 | 10.2 |
| GPT-o4-mini (OpenAI, 2025) | 59.6 | 26.9 | 33.4 | 13.2 | 46.5 | 20.1 |
| *Our Models* | | | | | | |
| Base Model (Bai et al., 2025) | 45.3 | 17.5 | 19.8 | 5.20 | 32.6 | 11.4 |
| CompassNav (SFT) | 54.1 | 23.1 | 23.9 | 7.90 | 39.0 | 15.5 |
| CompassNav (SFT+RFT) | **61.6** | **27.8** | **35.6** | **14.8** | **48.6** | **21.3** |

**Implementation Details.** CompassNav is built upon the open-source Qwen2.5-VL-7B (Bai et al., 2025) model. It is trained using our two-stage SFT-then-RFT recipe. All detailed configurations, including specific training frameworks, hyperparameters, and hardware, are deferred to Appendix E.

## 5.2 Main Results

To comprehensively evaluate CompassNav, we structure our comparisons in two parts. First, we benchmark against state-of-the-art modular methods, which represent the dominant approach in goal-driven navigation. Second, we compare with other end-to-end LVLMs to isolate the effectiveness of our training paradigm.

**Comparison with Modular Navigation Methods.** First, we benchmark our end-to-end CompassNav against state-of-the-art modular systems, with results presented in Table 1. These baselines employ complex, multi-stage pipelines, often relying on explicit memory like semantic maps (Yokoyama et al., 2024a) or history images. Instead of building an explicit world model, we focus on distilling spatial reasoning capabilities directly into the model's parameters. We show that our simpler agent achieves competitive, and often superior, performance, suggesting that a focus on the learning paradigm itself is a more direct and efficient path toward capable navigation agents.

**Comparison with End-to-End LVLMs.** To isolate the effectiveness of our framework from the base model's inherent capabilities, we conduct a comparison against other powerful LVLMs within the same end-to-end setting. As shown in Table 2, our CompassNav agent significantly outperforms

Table 3: Performance comparison on the NavNuances benchmark. We compare CompassNav against GPT-4 based agents and the base model to evaluate atomic reasoning capabilities.

| Method | DC | NU | LR | RR | VM |
|---|---|---|---|---|---|
| NavGPT-3.5 | 81.87 | 20.51 | 58.54 | 39.63 | 7.06 |
| NavGPT-4 | 91.87 | 34.78 | 54.83 | **67.61** | 11.36 |
| NavGPT-4V | **92.68** | **39.13** | **62.87** | 56.25 | 13.64 |
| Qwen2.5-VL-7B | 79.62 | 19.10 | 50.37 | 49.00 | 7.88 |
| CompassNav-7B | 82.54 | 23.85 | 59.92 | 59.31 | **22.94** |

these much larger, general-purpose models like GPT-4o (Hurst et al., 2024) and even surpasses o4-mini, a model renowned for its powerful general reasoning capabilities.

We conduct a special comparison with Nav-R1 (Liu et al., 2025b), a concurrent work that also fine-tunes an LVLM for embodied tasks. Nav-R1 aims to build a general-purpose agent and has achieved progress on multiple tasks. However, its training paradigm remains rooted in imitating a single expert path. On HM3D-OVON (Yokoyama et al., 2024b)benchmark, our method surpasses Nav-R1 on both key metrics (See Figure 6 for details). Notably, we achieve it while using one-tenth of the training data and starting from a general-purpose LVLM, rather than a 3D-specialized model. This result strongly suggests that our paradigm is a more efficient and effective path.

**Evaluation on Decision Understanding.** While standard metrics like SR and SPL measure navigation outcomes, they do not explicitly quantify the agent's spatial reasoning process. To validate our claim of decision understanding, we evaluate CompassNav on the NavNuances benchmark (Wang et al., 2024b), which disentangles navigation into atomic capabilities: Direction Change (DC) for basic turn instructions, Numerical Comprehension (NU) for counting and sequential memory (e.g., second door), Landmark Recognition (LR) for fine-grained visual grounding, Region Recognition (RR) for semantic room understanding, and Vertical Movement (VM) for multi-floor 3D spatial reasoning (e.g., "Go upstairs").

As shown in Table 3, CompassNav demonstrates decisive gains in metrics demanding deep spatial reasoning. Most notably, we observe a $\sim 3\times$ improvement over the base model in VM, significantly outperforming even NavGPT-4V. This confirms that our training paradigm successfully imbues the agent with an understanding of structural connectivity and 3D reasoning, going beyond simple path imitation. While CompassNav significantly improves upon the base model across all metrics, a performance gap remains compared to GPT-4V in DC and NU. This is largely due to task alignment and model scale: CompassNav is optimized for goal-oriented exploration rather than the rigid instruction following required by VLN (impacting DC). The lower NU score reflects a known limitation of 7B-scale models: they are more prone to hallucinations in memory-dependent long-context tasks compared to larger foundational models.

## 5.3 ABLATION STUDIES

**Efficacy of the SFT Stage.** Our experiments first validate the necessity of a proper policy initialization.

As show in Table 4, attempting RFT directly from this base model yields only a marginal 3.7 improvement in SR due to inefficient exploration from a naive policy.Applying RFT on this SFT-initialized model further improves performance by another 12.3, confirming the synergy of our two-stage approach.And recently, some methods only teach the model to output the action space of navigation tasks in the SFT stage. In difficult goal navigation tasks, this method actually deteriorates the effect.

Table 4: Efficacy of the SFT Stage.

| Config. | SR | SPL |
|---|---|---|
| base Model | 19.8 | 5.20 |
| SFT (Action only) | 17.9 | 5.78 |
| SFT (Full) | 23.3 | 7.90 |
| RFT (from Scratch) | 23.5 | 6.95 |
| RFT (from SFT) | **35.6** | **14.8** |

**Analysis of the Gap-Aware hybrid Reward Function.**The quantitative results in Table 5b show that our Reward significantly outperforms others that only consider absolute correctness or relative distance. A deeper analysis, shown in Figure 6, reveals the principles behind its effectiveness.Our reward function is designed to be adaptive: it should provide a strong, decisive signal in high-certainty scenarios while remaining nuanced to encourage exploration in low-certainty ones. Figure 6(left)

Table 5: Ablation studies on the components of our framework.

| Max Bouns | SR | SPL | Reward. | SR | SPL | Temperature | SR | SPL |
|---|---|---|---|---|---|---|---|---|
| B = 0.5 | 31.3 | 12.2 | Binary | 29.5 | 11.1 | T = 0.2 | 33.2 | 13.3 |
| B = 1.0 | **35.6** | **14.8** | Min-Max | 29.2 | 12.5 | T = 0.5 | **35.6** | **14.8** |
| B = 1.5 | 27.9 | 11.7 | Ours | **35.6** | **14.8** | T = 0.8 | 33.7 | 13.9 |
| (a) Efficacy of the Max Bonus | | | (b) Analysis of the Reward Func | | | (c) Efficacy of the Temperature | | |

visualizes the reward gap across different hyperparameters, our chosen parameters(T=0.5, B=1.0) achieve a strong gap in high-certainty cases while maintaining a moderate one in low-certainty cases, and as show in Table 5a and 5c, this combination also achieved the best results.

Furthermore, the training dynamics in Figure 6(right) illustrate a critical insight. While the Binary and Min-Max reward models achieve deceptively high scores during training, this merely indicates success at a simplistic proxy task: perfectly mimicking the single best action. In contrast, our Gap-Aware hybrid Reward, though yielding a lower absolute score, provides a more meaningful signal by teaching the model to evaluate all options. This fosters a more generalizable reasoning capability and demonstrates that a reward function's alignment with the true objective of robust navigation is more critical than the raw magnitude of its score.

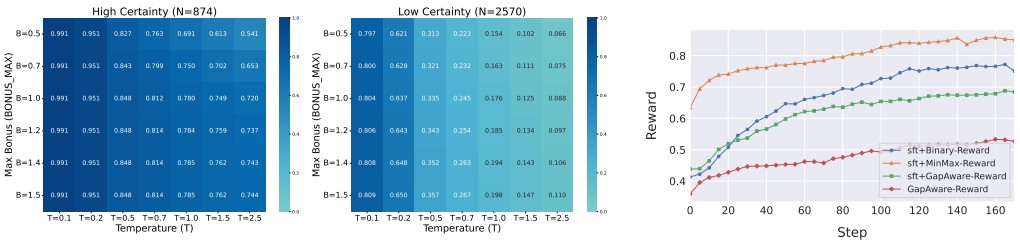

Figure 6: Left heatmaps showing the reward gap between the best and second-best actions under High and Low Certainty scenarios. Right training reward curves for different reward functions.

## 6 CONCLUSION

In this work, we present CompassNav, a framework that successfully moves beyond the prevailing paradigm of *Path Imitation* towards a more robust *Decision Understanding* approach in navigation. To enable this shift, we make two core contributions: the Compass-Data-22k dataset, which offers panoramic, per-action supervision far exceeding single-path imitation, and a novel Gap-Aware Hybrid Reward function that provides adaptive feedback tailored to decision certainty.Integrated into an SFT-then-RFT recipe, our framework transforms a 7B parameter LVLM into an expert navigator that establishes a new SOTA. It not only outperforms even larger proprietary models in simulation but also demonstrates robust performance in real-world deployment. Our findings paving the way for future research in low-cost, intelligent embodied agents.

### ETHICS STATEMENT

We adhere to the ICLR Code of Ethics. Our Compass-Data-22k dataset is generated via `habitat-sim` using public scenes (e.g., HM3D) without PII or human subjects. We distill knowledge from the open-source Qwen-QvQ to ensure licensing compliance. Acknowledging potential architectural biases in the source data, we commit to releasing all datasets and models to support transparent research in assistive robotics.

### REPRODUCIBILITY STATEMENT

To ensure full reproducibility, we release the Compass-Data-22k dataset, fine-tuned weights (LoRA), and all code for data generation, training, and physical deployment. Detailed hyperparameters and implementation specifics are documented in Appendix E.

## THE USE OF LARGE LANGUAGE MODELS (LLMS)

An LLM was used solely for refining text clarity and grammar. No AI tools were involved in research ideation, experimental design, or data analysis. The authors verify the accuracy of the final manuscript and assume full responsibility for its content.

## ACKNOWLEDGMENTS

This work is supported by the National Natural Science Foundation of China (No.62302167, 62476224, U23A20343, W2521174); Shanghai Committee of Science and Technology (No.25511103300, 25511104302, 25511102700); Young Elite Scientists Sponsorship Program by CAST YESS20240780;

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

## A   MORE RELATED WORK AND MORE EXPERIMETNS RESULTS

### A.1   MORE EXPERIMENTAL RESULTS

Based on current similar work, Nav-R1 was tested on other obejct navigation benchmarks HM3D-OVON (Yokoyama et al., 2024b). We also conducted tests with the same setting on the val unseen split of this benchmark. The results are shown in the table 6. Our method only used one tenth of their training data to train the original Qwen2.5-VL-7B, which still exceeded their model trained from 3D-R1. This further demonstrates the superiority of our method.

Table 6: Obejct Navigation results on HM3D-OVON val-unsenn

| Method | Val-Unseen | |
|---|---|---|
| | SR | SPL |
| Uni-NaVid  (Zhang et al., 2024) | 39.5 | 19.8 |
| MTU3D  (Zhu et al., 2025) | 40.8 | 12.1 |
| Nav-R1  (Liu et al., 2025b) | 42.2 | 20.1 |
| CompassNav (Ours) | **43.5** | **21.6** |

To address concerns regarding domain diversity and demonstrate robustness beyond photorealistic scans (HM3D/MP3D), we extended our evaluation to the AI2-THOR (ProcTHOR) dataset (Deitke et al., 2022). We utilized the Habitat-compatible version to evaluate on over 8,000 episodes in unseen synthetic environments. These environments feature distinct "game-like" textures and layouts, presenting a significant domain shift from our training data.

Table 7: Performance comparison on the AI2-THOR (ProcTHOR) benchmark. This evaluates Sim-to-Sim generalization from photorealistic training (HM3D) to synthetic testing environments.

| Method | SR | SPL |
|---|---|---|
| EmbCLIP | 47.0 | 20.0 |
| ProcTHOR  (Deitke et al., 2022) | 55.0 | 23.7 |
| CompassNav (Ours) | **58.1** | **32.8** |

As shown in Table 7, CompassNav significantly outperforms both the baseline (EmbCLIP) and the domain-specific ProcTHOR agent. Crucially, the massive improvement in SPL (+9.1%) indicates that our agent plans highly efficient paths even in unfamiliar visual domains. This strongly supports that our agent has learned transferable spatial logic rather than overfitting to specific visual styles.

**More Hyperparameter Analysis.** We further investigate the stability of our training recipe by analyzing two critical hyperparameters: the KL divergence strategy and the GRPO group sampling size.

Table 8: Hyperparameter sensitivity analysis on KL Divergence strategy and GRPO Sampling Size ($K$).

(a) Analysis of KL Strategy.

| Method | SR | SPL |
|---|---|---|
| Baseline (Base Model) | 45.3 | 17.5 |
| Adaptive KL | 61.5 | 26.0 |
| Fixed KL (Ours) | **61.6** | **27.8** |

(b) Ablation on Sampling Size $K$.

| Size | SR | SPL |
|---|---|---|
| $K = 3$ | 60.2 | 26.1 |
| $K = 5$ (Ours) | **61.6** | **27.8** |
| $K = 7$ | 61.8 | 26.0 |

As shown in Table 8a, we explored an adaptive KL mechanism ($kl\_target = 0.5, kl\_horizon = 2560.0$). However, this did not yield performance gains compared to our tuned Fixed KL. We hypothesize that in our specific GRPO setup combined with the Gap-Aware Reward, a fixed KL coefficient acts as a more stable anchor to the SFT policy. This effectively prevents the policy from drifting too far during the initial unstable exploration phase, ensuring consistent convergence. We also evaluated the robustness of the group size $K$ in the GRPO algorithm. As presented in Table 8b, performance is generally robust across values. We observe that $K = 5$ offers the optimal balance between computational efficiency and variance reduction.

## B    STANDARD BENCHMARKS AND TASKS

**Scene Datasets.**    Our experiments are primarily conducted on two large-scale, standard indoor navigation scene datasets: First, Habitat-Matterport 3D (HM3D) (Ramakrishnan et al., 2021b) is a large-scale dataset of 3D indoor environments, consisting of 1,000 high-resolution scans of residential and commercial spaces. Our work utilizes the version split for the Habitat Challenge, with the `train` split comprising 800 scenes and the `val` split containing 100 scenes. We report results on two semantic annotation versions: HM3Dv1 (Habitat Challenge 2022, using HM3D-Semantics-v0.1 (Ramakrishnan et al., 2021a)) and HM3Dv2 (Habitat Challenge 2023, using HM3D-Semantics-v0.2 (Yadav et al., 2023)).Seconde, Matterport3D (MP3D) (Chang et al., 2017) is another large-scale RGB-D dataset featuring 90 building-scale scenes, composed of 10,800 panoramic views from 194,400 images.

**Task Datasets.**    We evaluate our method on several goal-driven navigation task datasets: Object-Nav (ON) (Chaplot et al., 2020) requires agents to find an object of a given category. We evaluate on both the v1 version (2000 episodes, 20 scenes, 6 categories) and the v2 version (1000 episodes, 36 scenes, 6 categories) from the Habitat Challenge; HM3D-OVON (Yadav et al., 2023) is a large-scale open-vocabulary ObjectNav benchmark that significantly expands the semantic range, featuring over 15k instances across 379 distinct object categories; Instance Image-Goal Nav (IIN) (Krantz et al., 2022) requires agents to find a specific object instance. The goal categories, including "Chair", "Bed", "Toilet", "Couch", "Plant", and "TV", are consistent with the ObjectNav task.

**Task Definition.**    In our work, the agent receives a posed RGB-D video stream and must execute a new action $a \in \mathcal{A}$ upon receiving a new observation. A key distinction exists between our two main tasks: for **IIN**, the goal is a specific object instance provided as a goal image; for **ON**, the goal is an object category provided as text.

Our action space $\mathcal{A}$ consists of a discrete set of high-level actions represented as arrows rendered directly onto the agent's observation. A low-level controller translates the chosen arrow into a sequence of primitive simulator actions: `move_forward` (25cm), `turn_left` (30°), and `turn_right` (30°). An episode is considered successful if the agent executes the `stop` action when it is within a predefined radius 1-meter of the goal, in less than a maximum of 500 steps.

To handle the `stop` action, we are inspired by multi-agent collaborative frameworks. Instead of cluttering the primary agent's visual input with an additional stop command overlay, we employ a dedicated Stop Agent. This agent, a non-fine-tuned Qwen2.5-VL-7B (Bai et al., 2025) model, is prompted at each step to decide whether the goal is sufficiently in view to terminate the episode. This decouples the navigation and stopping decisions, simplifying the task for the primary agent. The prompt for the Stop Agent please refer D.2

## C    DATA GENERATION PIPELINE

### C.1    ACTION PROPOSAL MODULE (APM)

Following VLMNav (Goetting et al., 2024), we used the Action Proposal Module (APM), a critical component that translates raw depth sensor data into a discrete set of high-level, human-interpretable candidate actions for the LVLM to reason about. The process is designed to ensure actions are safe, visually distinct, and biased towards efficient exploration.

First, at each timestep, the APM leverages the depth image to compute the traversable area within the agent's field of view. Based on the furthest navigable distance for each angle $\theta$, an initial, dense set of candidate actions, $A_{\text{initial}}$, is generated. Simultaneously, we construct a 2D voxel map using accumulated depth and pose information, marking all observed areas within a 2-meter radius as explored.

To encourage systematic exploration and prevent redundant movements, we filter $A_{\text{initial}}$ based on the exploration map. For each action, we define an exploration indicator variable $e_i$:

$$e_i = \begin{cases} 1 & \text{if its destination region is unexplored} \\ 0 & \text{if its destination region is explored} \end{cases} \tag{7}$$

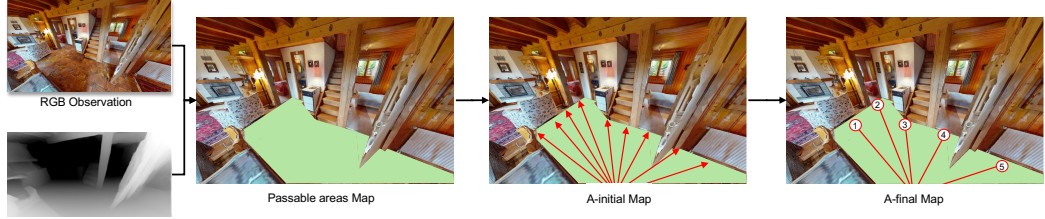

Figure 7: Action Proposal Module(APM) module workflow diagram

To build the final action set, $A_{\text{final}}$, we prioritize actions leading to new territory while ensuring sufficient visual spacing for the VLM to discern between options. This is achieved in a two-step filtering process. First, we greedily add unexplored actions ($e_i = 1$) that maintain a minimum angular spacing of $\theta_\delta$:

$$A_{\text{final}} \leftarrow A_{\text{final}} \cup \{(\theta_i, r_i) \mid e_i = 1 \text{ and } |\theta_i - \theta_j| \geq \theta_\delta, \forall (\theta_j, r_j) \in A_{\text{final}}\} \qquad (8)$$

To ensure the agent can still navigate within known areas and cover all directions, we then supplement $A_{\text{final}}$ with explored actions ($e_i = 0$), subject to a larger angular spacing $\theta_\Delta > \theta_\delta$:

$$A_{\text{final}} \leftarrow A_{\text{final}} \cup \{(\theta_i, r_i) \mid e_i = 0 \text{ and } |\theta_i - \theta_j| \geq \theta_\Delta, \forall (\theta_j, r_j) \in A_{\text{final}}\} \qquad (9)$$

For safety, the radius of each action is clipped via $r_i \leftarrow \min\left(\frac{2}{3} \cdot r_i, r_{\text{max}}\right)$. If no navigable actions are found, a 180° turn action is added to handle edge cases where the agent might be stuck. The final set of actions is then rendered as arrows onto the agent's RGB observation(Figure 7).

To bridge the gap between the LVLM's high-level decision-making and the simulator's low-level execution, we implement a deterministic low-level controller (Gu et al., 2025). This controller translates the agent's chosen high-level action, parameterized as a polar coordinate tuple $(r, \theta)$, into a discrete sequence of primitive actions executable within the Habitat simulator. The primitive action space consists of `move_forward(0.25m)`, `turn_left(30°)`, and `turn_right(30°)`.

The translation process follows a standard two-phase "turn-then-move" sequence to ensure predictable navigation outcomes:

1. **Rotation Phase:** The controller first addresses the angular component, $\theta$. The angle, given in radians, is converted to degrees. The number of required 30° turns is calculated using the ceiling function to ensure the agent turns at least the specified amount. The direction of the turn is determined by the sign of $\theta$: a positive value corresponds to `turn_left`, and a negative value corresponds to `turn_right`.

2. **Translation Phase:** After completing the rotation, the controller handles the distance component, $r$. The number of 0.25m forward steps is similarly calculated using the ceiling function. The agent then executes the `move_forward` action for the calculated number of steps.

This entire sequence of primitive actions is then executed by the simulator to complete the high-level action. Algorithm 1 provides a formal description of this process.

### C.2 DATA COLLECTION AND FILTERING

To generate our dataset, we employ a GenData Agent within the Habitat simulator. At each step, the agent receives an RGB-D observation, which is processed by the APM to generate candidate actions. For each action $(r, \theta)$, its global landing coordinates are calculated based on the agent's current pose. An oracle A* planner then computes the geodesic distance between each landing point and the goal's global coordinates.

The GenData Agent proceeds by selecting the action with the shortest distance. To collect a rich set of diverse yet effective trajectories, we implement a backtracking mechanism: if multiple actions have nearly equal A* distances, or if the decision certainty is below a threshold of 0.1, the agent records the current state (pose and candidate actions). After the current episode concludes, the GenData Agent returns to these recorded states to execute alternative choices, thus exploring and recording other viable paths to the same goal.

---

**Algorithm 1** High-level to Low-level Action Translation

---

1: **Input:** High-level action $A_{high} = (r, \theta)$
2: **Output:** A sequence of low-level actions $S_{low}$

3: **procedure** TRANSLATEACTION($A_{high}$)
4:     $S_{low} \leftarrow []$                                       ▷ Initialize an empty sequence
5:                                                       ▷ *Phase 1: Rotation*
6:     $total\_degrees \leftarrow \text{abs}(\theta \times 180/\pi)$
7:     $num\_turns \leftarrow \text{ceil}(total\_degrees/30)$
8:     **if** $\theta > 0$ **then**
9:         $turn\_action \leftarrow$ TURN_LEFT
10:     **else if** $\theta < 0$ **then**
11:         $turn\_action \leftarrow$ TURN_RIGHT
12:     **else**
13:         $num\_turns \leftarrow 0$
14:     **end if**
15:     **for** $i = 1$ **to** $num\_turns$ **do**
16:         Append $turn\_action$ to $S_{low}$
17:     **end for**
18:                                                    ▷ *Phase 2: Translation*
19:     $num\_forwards \leftarrow \text{ceil}(r/0.25)$
20:     **for** $i = 1$ **to** $num\_forwards$ **do**
21:         Append MOVE_FORWARD to $S_{low}$
22:     **end for**
23:
24:     **return** $S_{low}$
25: **end procedure**

---

While this pipeline could theoretically generate a very large dataset, we prioritize data quality and training efficiency, aligning with the "small data, more epochs" characteristic of RFT. We collected an initial set of 54k trajectories. This set was then rigorously filtered. We first used CLIP (Zhao et al., 2024) to discard trajectories where the goal image was blurry or semantically unclear (an occasional artifact of the simulator). Subsequently, we applied rule-based filtering to remove episodes where the agent was stuck in repetitive loops (e.g., constant turning). The final dataset consists of approximately 10k high-quality trajectories, a scale consistent with datasets commonly used for GRPO-based tasks.

**Challenges in Dense Annotation Generation.** This data generation process, although conceptually simple, faces high time costs. The computational cost of performing an A* search for *every* candidate at *every* timestep across thousands of trajectories is substantial. Our pipeline implements a highly parallelized, batched query system to the A* planner, making the generation of our large-scale corpus computationally tractable.

## D   DATASET STRUCTURE AND PROMPTS

### D.1   DATASET STRUCTURE

Our training corpus is composed of two distinct sets with differing distributions. The Supervised Fine-Tuning (SFT) set is distributed relatively evenly across six categories: chair (N=2,884), toilet (N=2,203), bed (N=1,697), tv screen (N=1,509), sofa (N=1,500), and plant (N=1,300).

In contrast, our Reinforcement Fine-Tuning (RFT) set is intentionally curated to reflect the natural long-tail distribution of objects found in the HM3D environment. This results in a deliberately imbalanced training set: chair (N=5,697), tv screen (N=2,790), plant (N=1,946), and toilet (N=1,100). This imbalance is not an artifact of our design but a characteristic of the benchmark itself, which is evident in the official validation set for Instance ImageNav (IIN) that exhibits a similar skew: chair (510), plant (155), bed (113), tv monitor (85), sofa (80), and toilet (57).

Given this inherent long-tail nature, evaluating on a similarly skewed test set could fail to penalize models that simply overfit to the most frequent categories. Therefore, to provide a more rigorous and unbiased assessment of our model's generalization capabilities, we conduct our primary evaluation on the ObjectNav validation set, which features a significantly more balanced distribution: chair (195), sofa (187), toilet (166), bed (165), plant (152), and tv monitor (135). This allows for a stricter test of the agent's ability to succeed across both common and rare object categories.

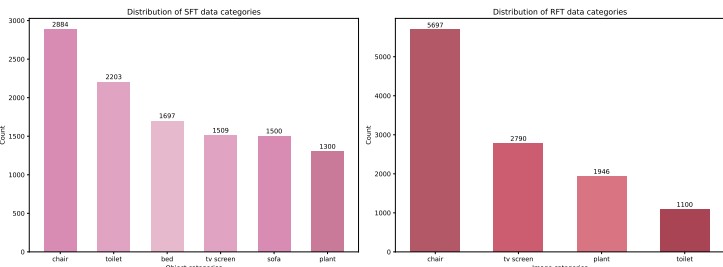

Figure 8: Category details of targets in SFT and RFT datasets

## D.2 PROMPTS

The following tables show the prompt templates used for the model in various navigation tasks.

Table 9: Prompt for CompassNav in the Object Navigation Task.

**SYSTEM PROMPT:**

You are an embodied robotic assistant, with an RGB image sensor. You observe the image and instructions. Given to you and output a textual response, which is converted into actions that physically move you. Within the environment. You cannot move through closed doors. "

**USER PROMPT:**

TASK: navigate to the nearest $\{goal.upper()\}$, and get as close to it as possible. Use your prior knowledge about where items are typically located within a home.

Current observation has $\{num_{actions} - 1\}$ potential actions shown as red arrows labeled with numbers in white circles. NOTE: choose action 0 if you want to turn around or dont see any good actions.

Critical Decision Framework:

1. Strategic decision (compose your plan internally, then select one action)
   - Where am I now(pose/heading)? What do I currently see (salient objects/structures/cues for goal.upper())
   - For each visible action, where will it likely take me next?"
   - For each visible action, estimate probability of encountering goal.upper() soon (based on priors, observed cues, spatial continuity, and last-seen evidence)

2. Action Selection
   - Output the all thinking process in <think></think> and the final answer in <answer></answer> tags.
   - Please strictly follow the format. and match potential actions to strategic decision.

3. CONSTRAINTS: Can go up and down stairs, but no closed doors.

Table 10: Prompt for Stop Agent in the Object Navigation Task.

**SYSTEM PROMPT:**

You are an expert visual analysis agent for a home navigation robot. Your mission is to determine if the robot has successfully reached its target object and should stop.The target object is: {goal.upper()}.

**USER PROMPT:**
The robot should stop ONLY if it is within 1 meter of the target object. This ensures it is close enough for any potential interaction. Additionally, it must be directly in front of the CORRECT object with a clear, unobstructed view.
CRITICAL INSTRUCTIONS

- Distinguish the target from visually similar objects. A CHAIR is not a SOFA, a SOFA is not a BED.
- Follow the reasoning steps strictly inside <think> tags.
- Provide your final command ONLY inside <answer> tags.
- Do NOT output anything outside <think> and <answer> tags.

Follow these steps in your reasoning process inside the <think> tags:

1. Identify Goal: State the target object you are looking for.
2. Scene Analysis: Briefly describe the main objects you see in the image.
3. Target Verification: Is the target object {goal.upper()} present in the image? Verify its identity against similar objects and state your confidence.
4. Proximity and Distance Assessment: If the target is present, you must evaluate if the robot is within 1 meter of it. Use visual cues to estimate this distance: Does the object appear very large and fill a significant portion of the image? Are fine details, like fabric texture or wood grain, clearly visible? Is the view completely unobstructed? The robot should only stop if it is confirmed to be within this 1-meter range.
5. Final Conclusion: Based on all the above, make a final decision on whether to stop or continue, and provide a brief justification.

Finaly, final Command (inside <answer>): Output ONLY 1 if the robot should STOP.Output ONLY 0 if the robot should CONTINUE.Do NOT output any other characters or words.

EXAMPLE: <think> 1.Identify Goal:The target is a SOFA. 2.Scene Analysis: I see a living room with a large grey sectional sofa, a coffee table, and a TV stand in the distance. 3.Target Verification: The large grey object is definitely a SOFA, not an armchair or a bed. I am highly confident. 4.Proximity and Distance Assessment: The sofa is very large in the frame,The fabric texture is clearly visible. Based on these cues, I estimate the robot is well within 1 meter of the sofa. The view is unobstructed. This is the correct stopping position. 5.Final Conclusion: The robot has successfully reached the SOFA and is positioned correctly within the 1-meter requirement. It should stop. </think> <answer>1 </answer>

Table 11: Prompt for CompassNav in the Instance Image-goal Navigation Task.

**SYSTEM PROMPT:**

You are an embodied robotic assistant, with an RGB image sensor. You observe the image and instructions. Given to you and output a textual response, which is converted into actions that physically move you. Within the environment. You cannot move through closed doors. "

**USER PROMPT:**
You can see a composite image composed of two independent images pieced together, separated by a white border:
The RIGHT image, labeled 'GOAL IMAGE', shows the TARGET OBJECT (the {goal['name']}) you need to find.

The LEFT image represents your CURRENT VIEW (what you are actually seeing now). Current observation has $\{num_{actions} - 1\}$ potential actions shown as red arrows labeled with numbers in white circles.Note: choose action 0 if you want to turn around or don't see any good actions, each arrow represents a potential action and is labeled with a number in a white circle indicating the location you would move to if you took that action. TASK:

1. Observe the right part. Identify the {goal['name']}) by focusing on intrinsic attributes like shape, color, and material, ignoring the environment surrounding the object.

2. Use your prior knowledge about where items are typically located within a home."

3. Describe what you see and any clues that could help you find the {goal['name']}).

4. Next, decide which direction you should go in.

5. You cannot go through closed doors. You cannot go up or down staircases. Ensure that the object being searched for is on the same floor.

Finally, choose the best action as your final answer.Output the thinking process in <think></think> and the final answer in <answer></answer> tags.

---

Table 12: Prompt for Stop Agent in the Instance Image-goal Navigation Task.

---

**SYSTEM PROMPT:**

You are an expert visual analysis agent for a home navigation robot. Your mission is to determine if your current view precisely matches the goal view, and if you should stop.

**USER PROMPT:**

You are given a composite image made of two parts:The LEFT image is your CURRENT EGO-VIEW.The RIGHT image is the GOAL IMAGE, showing the specific object instance you must find.You should command the robot to STOP only if the CURRENT VIEW contains the exact object from the GOAL IMAGE, and you are in an ideal position for interaction. This means you are directly in front of, and very close to, the target, with a clear, unobstructed view that matches the perspective of the GOAL IMAGE as closely as possible. CRITICAL INSTRUCTIONS

- The goal is a SPECIFIC INSTANCE, not just a category. A different chair, even of the same type, is NOT the target.

- Pay close attention to fine-grained details: color, shape, texture, and surrounding context to verify the instance match.

- Follow the reasoning steps strictly inside <think> tags.Provide your final command ONLY inside <answer> tags.Do NOT output anything outside <think> and <answer> tags.

Follow these steps in your reasoning process inside the <think> tags:

1. Analyze Goal Image (RIGHT): Briefly describe the target object instance in the GOAL IMAGE, noting its key visual features (e.g., 'a wooden armchair with blue cushions').

2. Analyze Current View (LEFT): Briefly describe the main objects you see in your CURRENT VIEW.

3. Instance Verification: Compare the objects in the CURRENT VIEW to the GOAL IMAGE. Is the specific target instance present? Justify your conclusion by comparing details like color, model, and wear/tear. State your confidence.

4. Proximity and Distance Assessment: If the target instance is present, evaluate your proximity and pose. Does the current view of the object match the scale and perspective of the goal image? Are you positioned for direct interaction?

5. Final Conclusion: Based on all the above, make a final decision on whether to stop or continue, and provide a brief justification.

Finaly, final Command (inside <answer>): Output ONLY 1 if the robot should STOP.Output ONLY 0 if the robot should CONTINUE.Do NOT output any other characters or words.

EXAMPLE: <think> 1.Analyze Goal Image (RIGHT): The goal image shows a specific wooden dining chair with a distinctive high back and a circular, light-colored seat cushion. 2.Analyze Current View (LEFT): I see a dining area in my current view. There are several chairs around a table. One of them is a high-backed wooden chair. 3.Instance Verification: Comparing the chair in the current view to the goal image, the wood grain, the high-back design, and the specific circular cushion match perfectly. It is the correct instance. I am highly confident. 4.Proximity and Pose Assessment: The chair in the current view is very close, taking up a large portion of the frame, similar to the goal image. I am facing it directly. The pose is ideal for stopping. 5.Final Conclusion: I have successfully reached the correct chair instance and am positioned correctly. I should command the robot to stop. </think> <answer>1 </answer>

## E    MODEL AND TRAINING DETAILS

### E.1    TRAINING CONFIGURATIONS

**Supervised Fine-Tuning (SFT) Details**    We conduct the Supervised Fine-Tuning (SFT) stage using the LLaMA-Factory framework. The training process was performed on 8 A800 GPUs and took approximately 12 hours to complete. The key configurations and hyperparameters are summarized below.We initialize our model from the `Qwen2.5-VL-7B-Instruct` checkpoint. The task is set up as a standard supervised fine-tuning problem, with a maximum sequence length (`cutoff_len`) of 4096 tokens.

**Fine-tuning Strategy.**    We employ Parameter-Efficient Fine-Tuning (PEFT) using the LoRA method to reduce computational overhead. The LoRA modules are applied to all projection layers (`q_proj, v_proj, k_proj, o_proj`) as well as the feed-forward layers (`gate_proj, up_proj, down_proj`). Key LoRA hyperparameters include a rank (`lora_rank`) of 64, an alpha (`lora_alpha`) of 128, and a dropout rate of 0.05.

**Training and Optimization.**    The model is trained for 3 epochs with a global batch size of 128 (8 GPUs $\times$ 8 per_device_batch_size $\times$ 2 gradient_accumulation_steps). We use the AdamW optimizer with a cosine learning rate scheduler, starting with a learning rate of $2 \times 10^{-4}$ and a weight decay of 0.01. The warmup ratio is set to 0.03 of the total training steps. For efficiency, we utilize DeepSpeed ZeRO Stage 2, BF16 mixed-precision training, and gradient checkpointing.

**Reinforcement Fine-Tuning (RFT) Details**    We conduct the Reinforcement Fine-Tuning (RFT) stage using EasyR1, a simplified version of the Verl framework. The training was performed on 8 NVIDIA H200 GPUs, with each model being trained for approximately 170 steps. Due to the computational demands of the rollout process, each training step took about 20 minutes.The actor model was initialized from the LoRA weights obtained during the SFT stage. We use the **GRPO** (`adv_estimator:grpo`) algorithm for policy optimization. A KL penalty (`kl_coef`) of $1 \times 10^{-2}$ is applied between the actor and the frozen reference policy to regularize the policy updates and prevent divergence.

**Optimization.**    We use the AdamW optimizer with a learning rate (`lr`) of $1 \times 10^{-6}$ and a weight decay of $1 \times 10^{-2}$. The global batch size for policy updates was set to 128, with a per-device micro-batch size of 4.

**Rollout and Sampling.**    During the experience generation (rollout) phase, we sample $N = 5$ distinct responses for each prompt from the training set, which is similar to the number of high-level actions generated by APM. The sampling process uses a high temperature of 1.0 and a top-p of 0.99 to encourage exploration. The rollout batch size is set to 512. For evaluation, the sampling is more deterministic, with $N = 1$ and a temperature of 0.5. Our custom Gap-Aware hybrid reward function, implemented as a Python script, is used to score the generated responses.

### E.2    JUSTIFICATION FOR A MEMORY-AGNOSTIC TRAINING APPROACH

We acknowledge that sophisticated memory is crucial for embodied navigation. The central debate, however, is not *whether* to use memory, but *how* it should be integrated with large-scale LVLMs. We identify two distinct integration philosophies:

First is the training-integrated memory approach, where historical context is directly "baked into" the model's input and weights during training. The second is the externally-interfaced memory approach, where complex memory systems (e.g., a SLAM-generated topological map) (Yin et al., 2025; Cao et al., 2024) operate as external modules that the agent can query at inference time. We argue that for powerful, pre-trained LVLMs, the latter is a more promising and architecturally sound direction. Complex memory systems like SLAM excel at providing structured, metric/topological knowledge, but there is currently no effective methodology to distill such a complex system into the weights of an LVLM via end-to-end training. They are best utilized as supplementary, external tools.

Based on this perspective, we focused our empirical validation on the first philosophy: training-integrated memory. We implemented and tested the two most popular paradigms currently used in the E2E navigation literature for this purpose: summarizing history as text (a-la OctoNav (Gao et al., 2025)) and concatenating historical images (a-la VLN-R1 (Qi et al., 2025)). As shown in Table 13, our experiments consistently found that both of these mainstream methods for integrating memory into the training process lead to a degradation in performance.

Table 13: Ablation studies on the impact of including historical information in the input prompt. using both the Qwen2.5-VL-7B and o4-mini models for the Object-Goal Navigation task on HM3Dv2

| Model | Method | SR | SPL |
|---|---|---|---|
| Qwen2.5-VL-7B | No history | 45.3 | 17.5 |
| Qwen2.5-VL-7B | With history - Text | 45.0↓0.3 | 13.5↓4.0 |
| Qwen2.5-VL-7B | With history - Image | 34.7↓10.6 | 10.9↓6.6 |

| Model | Method | SR | SPL |
|---|---|---|---|
| o4-mini | No history | 59.6 | 26.9 |
| o4-mini | With history - Text | 54.1↓5.5 | 26.3↓0.6 |

We attribute this failure to fundamental flaws in how these methods handle historical context. For text summaries, the process of transcribing visual history into language is inherently lossy. It forces rich, high-dimensional visual information through the narrow bottleneck of text, inevitably simplifying or omitting crucial details. Furthermore, any errors or biases from the VLM performing the transcription are then propagated forward, potentially providing the agent with a distorted or misleading memory. Conversely, for concatenated images, the challenge becomes one of temporal confusion. We hypothesize the model struggles to consistently distinguish the current, actionable observation from the passive historical frames, sometimes misinterpreting a past view as its present reality. This spatiotemporal misalignment can lead to critical errors in self-localization and immediate planning.

Therefore, our decision to adopt a memory-agnostic training approach is a deliberate and principled one. It is not an omission, but a rejection of the current, ineffective paradigms for training-integrated memory. By decoupling the core policy from a fixed memory format, we ensure that CompassNav is trained as a pure and robust decision-making core. This modular design makes our framework flexible and future-proof, allowing it to serve as a powerful foundation that can readily integrate with more advanced, externally-interfaced memory solutions as this important research direction evolves.

## F    REAL-WORLD DEPLOYMENT CONSIDERATIONS

To validate the practical applicability and sim-to-real transfer capabilities of our CompassNav agent, we successfully deployed it on a physical robotic platform. This section details the hardware, system architecture, and qualitative results from our real-world experiments.

**Hardware and System Architecture**    Our experiments were conducted on the ROSMASTER X3(Figure 9), a versatile mobile robot developed by Yahboom. It is equipped with Mecanum wheels that allow for omnidirectional movement, providing high agility in constrained indoor environments. For perception, the robot utilizes an Orbbec Astra Pro depth camera to capture RGB-D images, which serve as the primary visual input for our model. The robot operates on the ROS 2 (Robot Operating System) framework, facilitating robust communication and control.

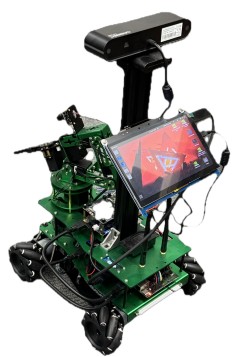

Figure 9: ROSMASTER X3

Our 7B model is computationally intensive, so our real-world deployment used a three-part system. This setup consisted of the ROSMASTER X3 robot, a local computer that acted as a relay, and a remote server with an NVIDIA H200 GPU to run the CompassNav model. All three devices communicated over the same Local Area Network (LAN). The process worked in a continuous loop: the robot sent its current camera view to the local computer, which combined the image with the task instruction (e.g., "find the trash can") and forwarded the pair to the server. On the server, our model processed the input and decided on a action command. This command was sent back to the local computer, which then translated it into low-level motor instructions for the robot to execute. This real-time setup allowed our model to act as the robot's brain and guide its navigation.

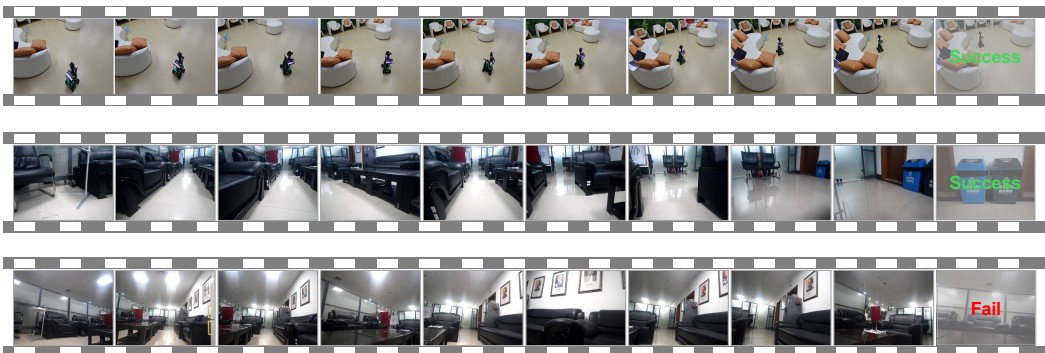

Figure 10: **Qualitative results from real-world deployment.** The first two rows show a successful navigation episode controlled by our CompassNav agent, presented from third-person and first-person perspectives, respectively. The agent successfully navigates a complex office environment to reach the "trash can". The third row shows a comparative trial using the closed-source model GPT-4o in a zero-shot setting.Despite its strong general capabilities, GPT-4o fails the task by colliding with an obstacle midway through the episode.

**Qualitative Sim-to-Real Results** We conducted several ObjectNav tasks in real-world indoor office environments. Figure 10 presents a compelling comparison of navigation episodes for the task "find the trash can".

As illustrated, our CompassNav agent, demonstrates robust navigation capabilities. The first two rows show that it can effectively process real-world visual information, make sensible decisions to maneuver around furniture, and successfully reach its goal. In contrast, the third row highlights the limitations of a general-purpose model GPT-4o (Hurst et al., 2024) for this embodied task, it struggled with spatial reasoning and obstacle avoidance, leading to task failure. These results strongly suggest that our proposed training framework not only excels in simulation but also produces a robust policy that successfully transfers to the complexities and visual nuances of the real world, validating the effectiveness of our approach.More detailed deployment videos can be viewed in our supporting materials.

# G Qualitative Analysis

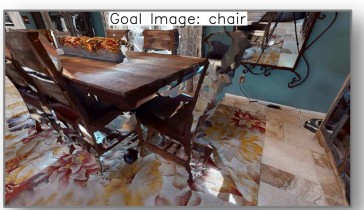

Goal Image: chair

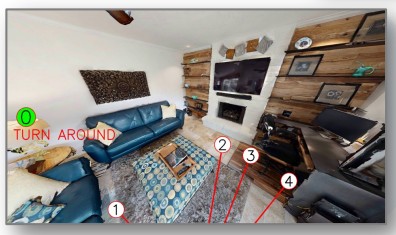 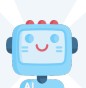

**Thought:** *The current view shows a living room area with no chairs in sight. I need to find the* **chair**, *which is visible in the goal image. The goal image shows* a dining table with four chairs around it (brown wooden chairs). *The chairs in the living room are either couches or an ottoman—not the target. The chairs are likely off to the right side of the living room* near the table in the dining area. *I should first move around the couches (left/dark blue couch, ottoman) around the living space and* look for an exit or hallway to the right that leads to the dining room. *The only listed action that will allow me to re-orient and navigate to that area is action 0 (turn around). Once I turn around, I can locate a path to the right of the living area where the dining room and chairs are likely located.*

**Action: 0**

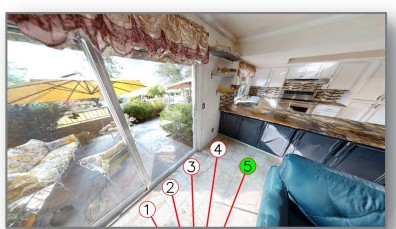 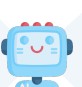

**Thought:** *The goal is to find the* **chair**. *In the current view, I see a glass sliding door to the outside, a closed white door slightly to the left, and a blue upholstered armchair just to the right of my viewpoint. I also see a bench in front of me and the interior kitchen beyond.* The closed white door would not be eligible. *There are five forward arrows: arrows 1 and 2 lead out through the glass door (no chairs outside), arrow 3 goes to the closed door,* arrow 5 take me to a hallway, *and arrow 4 goes toward the center of the kitchen floor.* In order to leave the living room and find the dining table, I choose action 5 to move toward and possibly closer to the chair.

**Action: 5**

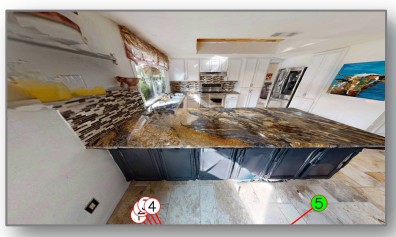 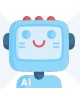

**Thought:** *I am in the kitchen facing an island.* On my right (arrow 5) I can move into the open dining area where I see dining chairs arranged around a table. *The goal is to find a* **chair**. *The chairs I can already see in the right image look like dining chairs and the left arrow 4 leads down a hallway or another room where I have not yet observed any chairs.* Therefore the best way to find the chairs visible in the right image is to move into the dining area via arrow 5.

**Action: 5**

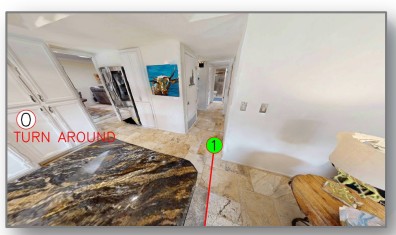 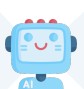

**Thought:** *My current view appears to be in a kitchen, looking towards a central hallway. The goal is to find a* **dining chair**, *which is typically located in a dining room or an open-plan dining area. The available actions are "0: TURN AROUND" and "1: MOVE FORWARD". Turning around (Action 0) would likely orient me back into the kitchen, which is a less probable location for a formal dining set. Action 1,* moving forward, leads down a main corridor which likely connects to the primary living and dining spaces of the house. *To find a new area like a dining room, proceeding along the main thoroughfare is the most logical exploratory step.* Therefore, moving forward (Action 1) is the best immediate action to advance towards the goal.

**Action: 1**

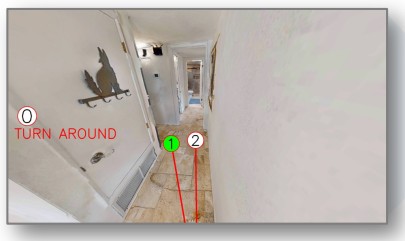 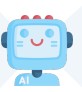

**Thought:** *- I am in a narrow hallway with no chairs in sight. There is two possible forward actions (1 and 2):* 1 continues down the hallway toward a larger room, *2 leads into a small doorway (likely a bathroom).The goal image shows a dining table with* **chairs**. *These chairs appear wooden, with cushions in dark and lighter shades. Bathrooms typically do not contain the same type of chair. If I turn around, I return to what I likely came from, but chairs could be behind me too, and it's an extra step.* Action 1 continues down the hallway toward a possible dining room. *Action 2 leads into a small bathroom – unlikely to have the chairs I see in the right image.* Thus, going forward via action 1 seems most promising to enter the room with visible chairs.

**Action: 1**

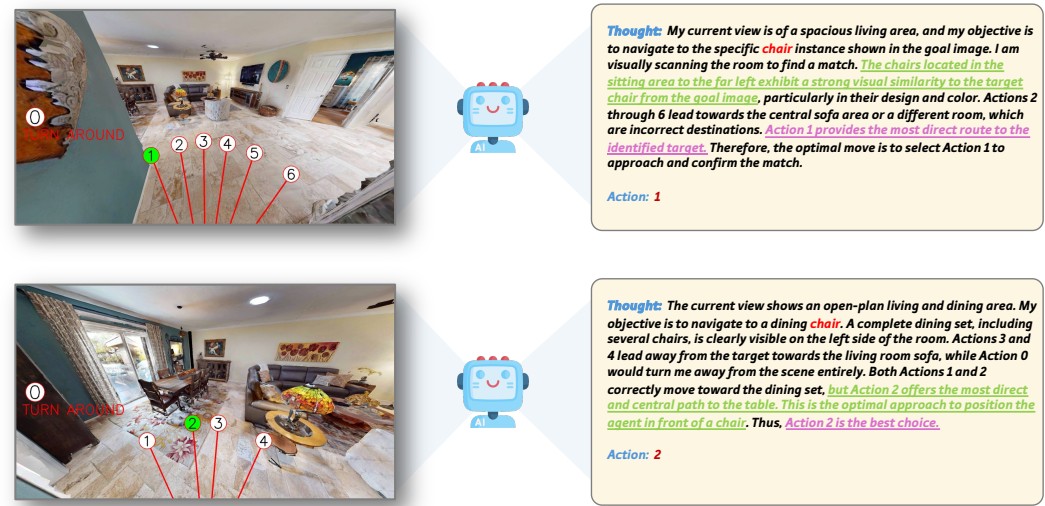

Figure 11: A qualitative, step-by-step visualization of our CompassNav agent. Task: find the chair in given image.

