# OpenReview forum: "CompassNav: Steering From Path Imitation to Decision Understanding In Navigation"
_ICLR.cc/2026/Conference — ICLR 2026 Poster_

### Official Review · Reviewer_iCB1 · 2025-10-27

**Soundness:** 3
**Presentation:** 3
**Contribution:** 2
**Rating:** 6
**Confidence:** 4

**Summary:**

This paper proposes CompassNav, a two-stage training framework that shifts the paradigm of embodied navigation from path imitation to decision understanding.
The authors argue that traditional imitation learning constrains embodied agents to replicate single expert trajectories, thereby limiting their exploration and generalization capability.
To address this, the paper introduces a large-scale dataset, Compass-Data-22k, which provides panoramic supervision signals by annotating all possible actions with A* geodesic distances. In addition, a gap-aware hybrid reward function is designed to modulate feedback strength based on decision certainty, which is then further integrated within a GRPO framework.
The resulting pipeline enables the 7B-parameter CompassNav model to develop an internal “compass,” allowing it to evaluate action quality beyond simple imitation.
Experimental results show that CompassNav achieves state-of-the-art performance on HM3Dv1/v2 and MP3D benchmarks and generalizes robustly to real-robot scenarios.

**Strengths:**

The paper makes a clear conceptual shift from imitation to understanding, reformulating navigation as a decision-making problem rather than a trajectory replication. This shift is appealing, helping to address the fundamental bottleneck of imitation-based LVLM training.
The Compass-Data-RFT-11k subset provides per-allocated A* annotations, offering richer training signals than existing single-path datasets. The data collection pipeline is transparent, reproducible, and logically linked to the proposed reward mechanism.
The gap-aware hybrid reward integrates the notions of distance-based preference and decision certainty in a simple but effective way. It adapts to ambiguous decision spaces and contributes to stable reinforcement fine-tuning.
The CompassNav agent consistently outperforms both modular pipelines (e.g., CogNav, UniGoal) and end-to-end LVLMs (e.g., GPT-4o, Nav-R1) despite using significantly fewer training samples. Results on real-world robotic navigation further support the performance.

**Weaknesses:**

*Conceptual ambiguity in “Decision Understanding”*

The paper repeatedly emphasizes “understanding” but does not quantitatively measure what constitutes understanding beyond improved reward signals. It remains unclear whether CompassNav truly learns transferable decision reasoning or merely fits dense reward distributions.

*Reliance on A\* as oracle*

While efficient, the A*-based distance labeling assumes deterministic environments and may not reflect realistic perception noise or dynamic obstacles. This could introduce bias into both reward computation and policy evaluation, especially in more practical environments.

*Limited diversity of evaluation environments*

Experiments focus mainly on HM3D variants and MP3D, which are visually and structurally similar. Broader generalization tests (e.g., Gibson, AI2-THOR, or out-of-domain tasks) would strengthen claims of decision-level robustness.

**Questions:**

The authors should explain more about the “decision understanding”. Are there any ways/metrics that help to demonstrate CompassNav’s reasoning rather than imitation bias?

In GRPO, the KL regularization term constrains policy updates relative to the SFT reference model. Do the authors experiment with adaptive or decayed KL coefficients to allow more flexible exploration during late training?

The RFT subset selectively explores “ambiguous nodes.” How is the sample balance ensured to prevent overrepresentation of uncertain scenarios in training?

---

> ### Author Response · Authors · 2025-11-21
>
> **Q1: Conceptual ambiguity in “Decision Understanding” (Response to Weakness1 & Question1)**
>
> **A1:** We agree that improved rewards alone are insufficient proofs. We address this concern through both conceptual clarification and  empirical verification (qualitative & quantitative).
>
> 1. **Conceptual Definition**: We define "Decision Understanding" as the shift from trajectory memorization to  explicit spatial reasoning . This means the agent must (a) internalize spatial structures and (b) execute a "Think-then-Act" chain—analyzing the state and estimating outcomes before acting.
> 2. **Qualitative Evidence** : We provide step-by-step visualizations of the navigation process in Appendix G (Figure 11). These qualitative results show the agent's explicit chain-of-thought: " Self-Localization-Action Projection- Value Estimation." This explicitly transparent reasoning chain confirms the model is actively evaluating logical links between actions and goals.
> 3. **Quantitative Verification**: To quantitatively measure if this reasoning is transferable, we adopted the NavNuances benchmark [1] (as also suggested by Reviewer Bopw). This benchmark decouples navigation into atomic reasoning capabilities. To save you time, we have summarized the core metrics of this benchmark for you:
>
>    * Direction Change (DC): Basic instruction following (e.g., "Turn left").
>    * Numerical Comprehension (NU): Counting and memory (e.g., "Enter the second room").
>    * Landmark Recognition (LR): Visual grounding (e.g., "Walk past the sofa").
>    * Region Recognition (RR): Semantic room understanding (e.g., "Go into the bedroom").
>    * Vertical Movement (VM): 3D reasoning (e.g., "Go upstairs").
>
>    Table 1: Comparison of performance on the NavNuances benchmark.
>
>    | Method        | DC    | NU    | LR    | RR    | VM    |
>    | :------------ | :---- | :---- | :---- | :---- | :---- |
>    | NavGPT3.5     | 81.87 | 20.51 | 58.54 | 39.63 | 7.06  |
>    | NavGPT4       | 91.87 | 34.78 | 54.83 | 67.61 | 11.36 |
>    | NavGPT4v      | 92.68 | 39.13 | 62.87 | 56.25 | 13.64 |
>    | Qwen2.5-VL-7B | 79.62 | 19.10 | 50.37 | 49.00 | 7.88  |
>    | CompassNav-7B | 82.54 | 23.85 | 59.92 | 59.31 | 22.94 |
>
>    As shown in  Table 1 , the results provide strong evidence for "Decision Understanding." CompassNav achieves decisive gains in metrics demanding deep spatial reasoning, most notably observing a ~**3x** improvement over the base model in Vertical Movement (VM) and significantly outperforming NavGPT-4v (22.94 vs. 13.64). This confirms that our new training paradigm successfully teaches the agent structural connectivity and visual grounding, enabling genuine decision-making beyond simple path imitation.
>
>    While CompassNav improves upon the base model across all metrics, performance gaps with GPT-4v in DC and NU remain due to task alignment and model constraints. CompassNav is trained for goal navigation exploration rather than the strict path adherence of VLN, which naturally impacts rigid instruction-following metrics like DC. Additionally, the precise View ID output required by the benchmark poses hallucination challenges for 7B models , and the NU gap reflects the known difficulty of long-horizon spatial memory in small LVLMs, a limitation we verified in our appendix  and identify as a key direction for future research.
>
>    [1] Wang, Z., Wu, M., Cao, Y., Ma, Y., Chen, M., & Tuytelaars, T. (2024). Navigating the nuances: A fine-grained evaluation of vision-language navigation.
>
> **Q2: Reliance on A as oracle(Response to Weakness 2)**
>
> **A2:** It is crucial to clarify that **$A^*$** distance is used  strictly as a training signal , never during inference. This follows the standard "Student-Teacher" paradigm: we use clean labels (**$A^*$** geodesic distance) to train a model that must operate on noisy inputs (RGB observations). The goal is for CompassNav to learn to *estimate* ideal geometry from visual cues. Our Real-World Deployment (Appendix F) empirically validates this: the agent successfully navigates dynamic, unmapped environments where no **$A^*$** oracle exists, proving it has internalized the navigation policy rather than relying on the oracle.

---

> ### Author Response · Authors · 2025-11-21
>
> **Q3: Limited diversity of evaluation environments (Response to Weakness 3)**
>
> **A3:** To addressing the concern about domain diversity and demonstrate robustness beyond photorealistic scans (HM3D/MP3D), we conducted a large-scale additional experiment on AI2-THOR (ProcTHOR). We utilized the Habitat-compatible version of ProcTHOR to test our model on 8,000+ episodes across unseen synthetic environments. This dataset features "game-like" textures and layouts distinct from our training data, serving as a rigorous test for Sim-to-Sim generalization. As shown in the table 2, CompassNav significantly outperforms both the baseline (EmbCLIP) and the domain-specific ProcTHOR agent.
>
> Table 2: Comparison of performance on the AI2-THOR benchmark.
>
> | Method     | SR         | SPL        |
> | :--------- | :--------- | :--------- |
> | EmbCLIP    | 47.0%      | 20.0%      |
> | ProcTHOR   | 55.0%      | 23.7%      |
> | CompassNav | **58.1%** | **32.8%** |
>
> Despite the significant domain gap, CompassNav outperforms the domain-specific ProcTHOR agent. Crucially, the **massive improvement in SPL (+9.1%)** indicates that our agent plans highly efficient paths even in unfamiliar visual domains, strongly supporting that it has learned transferable spatial logic.
>
>
> **Q4: Exploring Adaptive KL Divergence (Response to Question 2)**
>
> **A4:** Following your suggestion, we explored an adaptive KL mechanism ($kl\_target=0.5, kl\_horizon=2560.0$) and retrained the model for ~48 GPU-hours.
>
> Table 3: KL Divergence Ablation.
>
> | Method      | SR    | SPL   |
> | :---------- | :---- | :---- |
> | baseline    | 45.3% | 17.5% |
> | fixed KL    | 61.6% | 27.8% |
> | Adaptive KL | 61.5% | 26.0% |
>
> We found that Adaptive KL did not yield performance obvious gains compared to our tuned Fixed KL. We hypothesize that in our specific GRPO setup with the Gap-Aware Reward, a fixed KL coefficient provides a more stable anchor to the SFT policy, preventing the policy from drifting too far during the initial unstable exploration phase.
>
> **Q5: Sample Balance in RFT (Response to Question 3)**
>
> **A5:** We ensure sample balance by explicitly controlling the data distribution during dataset construction. Since we have the complete $A^*$ distances for every step, we first calculate a "certainty score" for all generated samples offline and categorize them into high, medium, and low certainty groups. Instead of using all explored ambiguous nodes, we enforce a fixed ratio for each group by downsampling the data. This guarantees that uncertain scenarios are included to encourage robust learning but are strictly limited to a specific proportion to prevent overrepresentation.
>
> We sincerely thank you for your insightful comments and constructive suggestions. We hope that our revisions and responses have addressed your concerns, and we would be happy to provide any further clarification if needed.

---

> ### Author Response · Authors · 2025-11-27
>
> Dear Reviewer iCB1,
>
> We would like to follow up to see if our response has satisfactorily addressed your concerns. We remain fully available to answer any further questions or provide additional clarifications if needed.

---

> > ### Comment · Reviewer_iCB1 · 2025-11-28
> >
> > Thank you for your detailed response. You have discussed/solved my comments thoroughly, where the additional experiments on NavNuances and ProcTHOR further enhance your claims. I would like to raise my score to strong accept.

---

### Official Review · Reviewer_Bopw · 2025-10-29

**Soundness:** 2
**Presentation:** 3
**Contribution:** 2
**Rating:** 4
**Confidence:** 3

**Summary:**

This paper proposes CompassNav, a paradigm for training large vision-language models (LVLMs) in navigation that shifts from Path Imitation to Decision Understanding. Instead of merely replicating expert trajectories, the authors aim to build agents that reason about the relative quality of alternative actions. To achieve this, the authors introduce compass-data-22k, a large-scale dataset that annotates every feasible action using A* geodesic distances, and design a gap-aware hybrid reward that dynamically adjusts feedback according to decision certainty. The training pipeline consists of  supervised fine-tuning stage and a reinforcement fine-tuning stage. Experiments on standard navigation benchmarks (HM3D, MP3D) show that CompassNav outperforms modular and end-to-end baselines, achieving new state-of-the-art results in object  and image-goal navigation.

**Strengths:**

The paper articulates a clear paradigm shift from trajectory imitation to decision-space reasoning. This reframing is conceptually strong.

The two-stage SFT-RFT training pipeline is methodically designed and empirically justified, overcoming the cold-start issue commonly faced in reinforcement fine-tuning.

**Weaknesses:**

While the paper claims to enable 'decision understanding',  the experiments still rely on conventional navigation metrics (SR, SPL). There is no explicit evaluation of whether the model actually understands or reasons about its decisions.  Some experiments similar to basic instruction-following tests could help the paper illustrate this issue, such as the tests conducted in [1]. Meanwhile, [1] involves VLM-based indoor navigation. Related comparisons or discussions are strongly recommended.


[1] Wang, Z., Wu, M., Cao, Y., Ma, Y., Chen, M., & Tuytelaars, T. (2024). Navigating the nuances: A fine-grained evaluation of vision-language navigation.

**Questions:**

What is the inference speed of this approach?

---

> ### Author Response · Authors · 2025-11-21
>
> Thank you for acknowledging the novelty of our decision-space reasoning. Below, we provide the specific evaluations on reasoning capabilities and inference speed.
>
> **Q1: Explicit reasoning evaluation**
>
> **A1:** We evaluated CompassNav on the  NavNuances benchmark [1] , which assesses atomic capabilities essential for navigation.This benchmark breaks down navigation into specific capabilities::
>
> * Direction Change (DC): Basic instruction following (e.g., "Turn left").
> * Numerical Comprehension (NU): Counting and memory (e.g., "Enter the second room").
> * Landmark Recognition (LR): Visual grounding (e.g., "Walk past the sofa").
> * Region Recognition (RR): Semantic room understanding (e.g., "Go into the bedroom").
> * Vertical Movement (VM): 3D reasoning (e.g., "Go upstairs").
>
> Table 1: Comparison of performance on the NavNuances benchmark.
>
> | Method        | DC    | NU    | LR    | RR    | VM    |
> | :------------ | :---- | :---- | :---- | :---- | :---- |
> | NavGPT3.5     | 81.87 | 20.51 | 58.54 | 39.63 | 7.06  |
> | NavGPT4       | 91.87 | 34.78 | 54.83 | 67.61 | 11.36 |
> | NavGPT4v      | 92.68 | 39.13 | 62.87 | 56.25 | 13.64 |
> | Qwen2.5-VL-7B | 79.62 | 19.10 | 50.37 | 49.00 | 7.88  |
> | CompassNav-7B | 82.54 | 23.85 | 59.92 | 59.31 | 22.94 |
>
> As shown in  Table 1 , the results provide strong evidence for "Decision Understanding." CompassNav achieves decisive gains in metrics demanding deep spatial reasoning, most notably observing a ~**3x** improvement over the base model in Vertical Movement (VM) and significantly outperforming NavGPT-4v (22.94 vs. 13.64). This confirms that our new training paradigm successfully teaches the agent structural connectivity and visual grounding, enabling genuine decision-making beyond simple path imitation.
>
> While CompassNav improves upon the base model across all metrics, performance gaps with GPT-4v in DC and NU remain due to task alignment and model constraints. CompassNav is trained for goal navigation exploration rather than the strict path adherence of VLN, which naturally impacts rigid instruction-following metrics like DC. Additionally, the precise View ID output required by the benchmark poses hallucination challenges for 7B models , and the NU gap reflects the known difficulty of long-horizon spatial memory in small LVLMs, a limitation we verified in our appendix  and identify as a key direction for future research.
>
> We also provide step-by-step visualizations of the navigation process in Appendix G (Figure 11). These qualitative results show the agent's explicit chain-of-thought: " *Self-Localization-Action Projection-* Value Estimation." This explicitly transparent reasoning chain confirms the model is actively evaluating logical links between actions and goals.
>
> [1] Wang, Z., Wu, M., Cao, Y., Ma, Y., Chen, M., & Tuytelaars, T. (2024). Navigating the nuances: A fine-grained evaluation of vision-language navigation.
>
> **Q2: Inference speed**
>
> **A1:** We evaluated CompassNav on an NVIDIA H200 GPU, matching our real-world deployment setup. The average end-to-end latency is approximately  **300ms per step**. In real-world experiments, the bottleneck is rarely the model inference; the robot's physical execution (e.g., moving 0.25m) takes significantly longer than 300ms. Thus, our model's latency fits comfortably within the control loop, enabling smooth, real-time navigation as demonstrated in our supplementary demo videos.
>
> We hope our responses address your concerns. If you have any further questions, please feel free to contact us. Thank you again for spending time reviewing our work and providing valuable feedback.

---

> ### Author Response · Authors · 2025-11-27
>
> Dear Reviewer Bopw,
>
> We would like to follow up to see if our response has satisfactorily addressed your concerns. We remain fully available to answer any further questions or provide additional clarifications if needed.

---

> > ### Comment · Reviewer_Bopw · 2025-11-28
> >
> > Thank you to the authors for providing more comprehensive experiments to analyze the performance. Under the same parameter budget, the proposed method demonstrates improved navigation understanding capabilities. I will therefore be raising my score.

---

### Official Review · Reviewer_6mHY · 2025-10-30

**Soundness:** 2
**Presentation:** 3
**Contribution:** 3
**Rating:** 4
**Confidence:** 3

**Summary:**

In this work, the authors address the object goal navigation problem. They introduce Compass-Data-22k, a 22k-trajectory dataset comprising 11k reasoning traces with multi-modal reasoning and goal selection for supervised fine-tuning (SFT) and 11k densely annotated reward samples for reinforcement fine-tuning (RFT). Actions are generated through an action proposal module, with rewards computed based on A* geodesic distances. The authors train the CompassNav agent using an SFT–RFT pipeline and propose a gap-aware hybrid reward function that encourages exploration in ambiguous cases and delivers decisive feedback when a clear optimal action exists. The resulting model achieves state-of-the-art performance on goal navigation benchmarks.

**Strengths:**

- The authors commit to open-sourcing both the dataset and codebase, which enhances reproducibility and can meaningfully benefit the research community.

- The proposed Compass-Data-22k dataset is well-structured and provides dense annotations that can facilitate broader research in embodied navigation and decision modeling.

- The gap-aware hybrid reward function is an interesting contribution, as it attempts to balance decisiveness and exploration through adaptive reward modulation.

- The inclusion of real-world robot deployment results complements the simulation benchmarks and strengthens the practical relevance of the work.

**Weaknesses:**

- Although the authors claim state-of-the-art performance, the improvements over baselines are marginal and may not clearly justify the methodological complexity.

- The claim that the model achieves “decision understanding” and this paper introduces “a new paradigm” seems to be overstated.
    - The reasoning traces used for SFT are synthetically generated by another multimodal LLM and may not reflect causal or grounded reasoning. Furthermore, since no contextual history is preserved between steps, the reasoning is effectively discarded in subsequent decisions. For instance, in Figure 2(3), the generated rationale—“I want to go inside the kitchen…”—would be forgotten by the next invocation, leaving the robot unaware of why it entered the kitchen in the first place. As a result, these generated explanations appear arbitrary and disconnected from actual decision-making, contributing little to “decision understanding.”
    - During RFT, the reasoning outputs are not incorporated into the reward computation, suggesting that the model’s reasoning is superficial rather than functionally integrated. As such, the method does not have much difference from standard robot learning paradigms (e.g., pretraining and finetuning using reinforcement learning with dense rewards).

- The motivation and necessity for the use of LVLMs are not fully convincing. Given access to depth and semantic information, classical navigation pipelines (e.g., SLAM + frontier exploration + object recognition) could potentially solve the same problem efficiently without large-scale data or training. Why use LVLMs for such tasks?

**Questions:**

- Regarding the gap-aware hybrid reward (line 304), what exactly do *best* $d_t^{(1)}$ and *second-best* $d_t^{(2)}$ mean? Are they derived purely from A* distances (i.e., shortest path to goal)? Or based on the model’s predicted actions (best corresponds to the most likely action output) and their corresponding A* distances?

---

> ### Author Response · Authors · 2025-11-21
>
> We thank the reviewer for recognizing the value of our paper. Below, we give our point-by-point responses to your comments:
>
> **General Response : Positioning in the Navigation Landscape**
>
> To better position our work within the landscape outlined in Figure 2, we distinguish our approach from traditional modular systems (e.g., SLAM). While modular methods provide robust geometric mapping, they often lack the semantic understanding required to follow complex linguistic instructions.
>
> Our work focuses on the End-to-End LVLM paradigm. Instead of relying on geometric maps, we utilize the inherent world knowledge of VLMs to guide navigation through  **semantic reasoning** . Our primary objective is to **embed** navigation capabilities directly into the generalist model's weights. We believe this shift is critical for building future agents that can reason semantically and interact seamlessly with humans, without being tethered to external engineered maps.
>
> **Q1:  Necessity of LVLMs（Response to Weakness 3）**
>
> **A1:** While classical pipelines (e.g., SLAM + Frontier Exploration) effectively answer the geometric question ("Can I go there?"), they lack the semantic intuition to answer ("Should I go there?"). This limitation is critical for two reasons:
>
> 1. Efficiency: A classical agent searching for a "toilet" might waste time mapping the entire living room. An LVLM-based agent uses common sense to recognize the room type and immediately head towards hallways or doors, pruning the search space.
> 2. Abstract Instructions: Classical pipelines struggle with abstract commands like "I need to wash my hands." LVLMs can parse this intent. Our work focuses on training this core semantic reasoning module, which is the foundation for more intelligent interaction.
>
> **Q2:  Assessment of Improvements（Response to Weakness 1）**
>
> **A2:** We believe the perception of "marginal" gains stems from comparing our learned End-to-End (E2E) method against highly engineered modular systems (Table 1). Modular systems currently utilize explicit, engineered memory (like 3D voxel maps) which naturally avoids geometric forgetting. However, our objective is to solve the harder problem of *learning* navigation policies without external aids. When evaluated strictly within the E2E paradigm (Table 2), *CompassNav* represents a significant step forward. We achieve a **+36.8% performance leap** over the base model and outperform much larger proprietary models like GPT-o4-mini. These results confirm that our "Decision Understanding" methodology effectively bridges the gap between open-source models and top-tier systems. We also openly discuss the current limitations of memory in E2E systems in Appendix E.2, which remains a key focus for our future work.
>
> **Q3:Validity of Decision Understanding  (Response to Weakness 2)**
>
> **A3:** We clarify the roles of SFT and RFT to explain why our reasoning is neither discarded nor superficial:
>
> * SFT: We need to clarify that the goal of SFT is not to memorize history, but to distill the "Think-then-Act" heuristic. By imitating how a teacher analyzes a single frame (e.g., "I see a hallway, so I move forward"), the model learns generalizable spatial reasoning. This allows the agent to make optimal decisions based on current observations, even without an explicit memory buffer.
> * RFT: In our GRPO framework, reasoning is functionally integrated. The policy generates reasoning tokens **$r$**, which conditionally produce the action **$a$** via **$P(a|r, obs)$**. Since gradients propagate from the reward back through **$a$** to **$r$**, the model is mathematically incentivized to generate valid reasoning. If the reasoning were arbitrary or hallucinated, it would fail to guide the correct action, resulting in low rewards. Thus, the reasoning is optimized to be the valid precursor to the decision.
>
> **Q4: Regarding Gap-Aware Reward ($d_t^{(1)}$ and $d_t^{(2)}$)**
>
> **A4:** We confirm that **$d_t^{(1)}$** and **$d_t^{(2)}$** are calculated strictly using Oracle **$A^*$** geodesic distances. They are ground-truth geometric measurements and are entirely independent of the model's predicted action probabilities.
>
> We hope our responses address your concerns. If you have any further questions, please feel free to contact us.

---

> ### Author Response · Authors · 2025-11-27
>
> Dear Reviewer 6mHY,
>
> We would like to follow up to see if our response has satisfactorily addressed your concerns. We remain fully available to answer any further questions or provide additional clarifications if needed.

---

### Official Review · Reviewer_oiRX · 2025-10-31

**Soundness:** 2
**Presentation:** 3
**Contribution:** 2
**Rating:** 4
**Confidence:** 4

**Summary:**

The paper proposes a vision-language navigation paradigm that shifts from traditional single-path imitation to understanding and modeling the relative value of all candidate actions. The authors built the Compass-Data-22k dataset based on the HM3D environment and designed a gap-aware hybrid reward mechanism that uses a softmax formulation to represent action quality, introducing a dynamic bonus under high-certainty conditions. The model, built upon Qwen2.5-VL-7B, employs a two-stage training strategy of SFT pretraining followed by RFT reinforcement to achieve reasoning-to-action policy learning. Evaluations on the HM3D and MP3D datasets for ObjectNav and ImageGoal tasks.

**Strengths:**

- A self-constructed Compass-Data-22k dataset provides dense annotations to support reward modeling.
- The gap-aware reward dynamically adapts to different uncertainty scenarios, enhancing training stability and exploration capability.

**Weaknesses:**

- The work mainly combines existing components—standard LVLM architecture and RLHF-style two-stage training—without introducing fundamentally new modeling or optimization techniques. The “gap-aware hybrid reward” is a reasonable design but conceptually close to prior contrastive or entropy-regularized rewards.
- The two-stage SFT–RFT pipeline and dense A*-based labeling increase training cost and reduce scalability compared to simpler imitation or zero-shot methods.
- Despite stronger reasoning, CompassNav underperforms recent zero-shot approaches like BeliefMapNav, indicating weaker generalization efficiency.
[1] BeliefMapNav: 3D Voxel-Based Belief Map for Zero-Shot Object Navigation
- In the navigation task, the determination of whether the agent has “reached the target” is delegated to an independent large language–vision model (such as Qwen2.5-VL-7B), serving as the Stop Agent. However, large language models are often sensitive to semantic thresholds and contextual ambiguities. As a result, the Stop Agent may prematurely trigger a stop when the target is only partially visible, semantically ambiguous, or occluded, leading to an overestimation of the Success Rate (SR). Therefore, the currently reported SR metric may not accurately reflect the true performance of the navigation system. It is necessary to additionally provide the Stop Agent’s independent accuracy, recall, and SR sensitivity analysis, or to describe the types of misjudgments that occur under occlusion and complex semantic target conditions.
- The paper proposes a complex framework (including RFT, SFT, reward module, candidate action generation, Stop Agent, and Bonus mechanism), but does not individually verify the necessity of each component. It lacks controlled experiments for key variables such as the number of candidate actions, angular resolution, reward function form, teacher model quality, and sampling strategy. In addition, statistical significance tests could be added to demonstrate the true stability of the performance improvements.

**Questions:**

- The paper lacks sufficient implementation details to ensure reproducibility — for example, the exact hyperparameters, model initialization, and architecture adaptation between CLIP and Qwen-7B are not clearly described. Could the authors clarify these details and release training configurations or code references?
- The proposed two-stage SFT–RFT training involves GRPO optimization and A*-based dense labeling, which appear computationally expensive. Could the authors provide quantitative analysis of training cost (e.g., GPU hours, sample efficiency, or convergence comparison) relative to existing navigation methods?
- Will the Compass-Data-22k dataset be publicly released? If so, could the authors provide more details about its annotation process and data generation pipeline?
- How significantly do the number of candidate actions and sampling parameters affect model performance? Have the authors conducted any sensitivity analysis or ablation studies to evaluate this?

---

> ### Author Response · Authors · 2025-11-21
>
> **Q1: Response to Weakness 1 (Novelty & Gap-Aware Reward vs. Entropy Regularization)**
>
> **A1:** We argue that our contribution lies in a fundamental modeling paradigm shift, reframing navigation from sequence imitation to state-based "Decision Understanding"—rather than architecture modifications.For optimization technique, We introduce Masked Multiple-Choice Decoding in the SFT stage (Eq. 1), which mathematically constrains output logits to valid action indices, effectively eliminating model hallucination.
>
> Regarding the  Gap-Aware Hybrid Reward , we clarify that it is conceptually distinct from—and in some aspects opposite to—entropy regularization. Entropy regularization typically encourages high entropy to promote random exploration. In contrast, our reward aims to teach the model  knowing when to be decisive and when to explore. In scenarios with a clear optimal solution (where the gap between the best and second-best action is large), our reward assigns a significant bonus to the optimal action, explicitly encouraging a  **low-entropy,** decisive policy. This dynamic adjustment based on environmental certainty is a domain-specific innovation tailored for navigation, fundamentally differing from generic regularization techniques.
>
> **Q2: Response to Weakness 2 (Cost & Scalability)**
>
> **A2:** We argue that our approach offers a superior trade-off between cost and performance compared to the alternatives. Zero-shot approaches rely on costly API calls to closed-source models like GPT-4o for every single inference step. In contrast, CompassNav incurs a one-time offline training cost, resulting in a model that requires only local GPU resources for inference with zero recurring API costs.
>
> Regarding scalability, our training utilizes GRPO with KL-divergence constraints to ensure the model does not deviate destructively from the base distribution. This preserves the general capabilities of the 7B model, ensuring that CompassNav maintains the same architectural scalability as standard LVLMs.
>
> **Q3: Response to Weakness 3 (Generalization vs. BeliefMapNav)**
>
> **A3:** While we acknowledge *BeliefMapNav* as a strong modular baseline, implying CompassNav has "weaker generalization" based solely on simulation metrics overlooks the critical distinction between End-to-End Learning and Modular Map-Based Engineering. We argue CompassNav offers superior practical generalization for three reasons:
>
> 1. Regarding real-world robustness, modular systems like *BeliefMapNav* rely on explicit 3D voxel grids that depend heavily on precise pose estimation, often suffering from degradation in the real world due to sensor noise and drift. Conversely, *CompassNav* demonstrates successful real-world deployment (Appendix F), confirming our learned policy is robust to physical dynamics.
> 2. Second, *BeliefMapNav* generalizes via engineered geometric consistency, whereas *CompassNav* generalizes via internalized semantic reasoning. While the latter is a harder learning task, it is essential for future generalist agents.
> 3. Finally, as an E2E LVLM, *CompassNav* retains linguistic capabilities, offering a scalable path toward robots that understand nuanced intent beyond simple object categories.
>
> **Q4: Response to Weakness 4 (Stop Agent Reliability & SR Overestimation)**
>
> **A4:** We first clarify that the Success Rate (SR) metric is strictly determined by the simulator based on the ground-truth geodesic distance (<1.0m); any premature stop is automatically penalized as a failure. To empirically verify the Stop Agent's reliability, we conducted a standalone evaluation on 1,000 balanced samples from the validation set (we select images that with a distance of>1.5m from the target object as negative samples; Select images that are less than 1.0m away from the target object and within the field of view as postive)
>
> Table 1: The confusion matrix.
>
> | Metric              | Value | Calculation                      |
> | :------------------ | :---- | :------------------------------- |
> | True Positive (TP)  | 347   | Distance<1m, Agent says stop     |
> | False Positive (FP) | 46    | Distance>1m, Agent says stop     |
> | False Negative (FN) | 153   | Distance<1m, Agent says continue |
> | True Negative (TN)  | 454   | Distance>1m, Agent says continue |
>
> The resulting confusion matrix (Table 1) reveals a Precision of 88.3% and a Recall of 69.4%. Crucially, the error analysis shows that False Negatives (153) significantly outnumber False Positives (46). This indicates that the agent is far more prone to missing the target (continuing to walk) than stopping prematurely. Consequently, the Stop Agent's errors primarily penalize our performance by extending path lengths or causing timeouts, confirming that our reported SR represents a reliable assessment of the navigation system's capabilities.

---

> > ### Author Response · Authors · 2025-11-21
> >
> > **Q5: Response to Weakness 5 & Question 4 (Ablation Studies & Significance)**
> >
> > **A5:** We have conducted extensive ablation studies of our framework's components, providing detailed analysis for the training stages (SFT vs. RFT) and the Reward Mechanism (Binary vs. Min-Max vs. Gap-Aware) in Tables 3 & 4. We utilized Qwen-QvQ as our teacher model as it represents the SOTA in open-source visual reasoning, avoiding proprietary models to ensure compliance. We also added an ablation on the GRPO sampling size (Table 2), finding that performance is robust across values, with **$K=5$** offering the optimal balance.
> >
> > Table 2:GRPO Sampling Size Ablation.
> >
> > | Method | SR    | SPL   |
> > | :----- | :---- | :---- |
> > | K=3    | 60.2% | 26.1% |
> > | K=5    | 61.6% | 27.8% |
> > | K=7    | 61.8% | 26.0% |
> >
> > For engineering parameters like angular resolution, we strictly adopted the default settings from  VLMNav [1]. Regarding the number of candidate actions, we clarify that this is not a fixed hyperparameter but is dynamically determined by the APM based on real-time depth and traversability analysis (Appendix C.1). Finally, given the massive performance margin (+36.8% over the base model and +6.2% over GPT-4o), the improvements significantly exceed typical random variance, demonstrating the robustness of our approach.
> >
> > [1] Goetting, D., Singh, H. G., & Loquercio, A. (2024). End-to-end navigation with vision language models: Transforming spatial reasoning into question-answering.
> >
> > **Q6: Response to Q1 (Implementation Details)**
> >
> > **A6:** We apologize for any ambiguity regarding the implementation details. We clarify that all specific hyperparameters and configurations are comprehensively listed in Appendix E. Regarding the model architecture, we emphasize that CompassNav does not utilize CLIP; instead, it employs the native Vision Transformer of Qwen2.5-VL as the visual encoder. The reference to CLIP in Appendix C.2 pertains exclusively to an offline data filtering step employed to remove blurry images from the training dataset. To ensure full reproducibility, we have provided the complete training codebase, including all configuration files, in the supplementary materials.
> >
> > **Q7: Response to Q2 (Computational Cost)**
> >
> > **A7:** Our method demonstrates high efficiency compared to traditional alternatives. The entire training pipeline requires less than 3 days on a standard 8-GPU node (approximately 12 hours for SFT on 8xA800 and 56 hours for RFT on 8xH200). This is orders of magnitude faster than standard Online RL approaches (e.g., DD-PPO), which typically demand weeks of training and millions of interaction steps. Finally, regarding the data generation overhead, the $A^*$ labeling represents a one-time, parallelizable(within 3 hours), and reusable offline cost that does not impede training scalability.
> >
> > **Q8: Response to Q3 (Dataset Release)**
> >
> > **A8:** Yes, the Compass-Data-22k dataset will be publicly released. Detailed descriptions of the annotation process and data generation pipeline are available in Section 3 and Appendix C.

---

> ### Author Response · Authors · 2025-11-27
>
> Dear Reviewer oiRX,
>
> We would like to follow up to see if our response has satisfactorily addressed your concerns. We remain fully available to answer any further questions or provide additional clarifications if needed.

---

> > ### Comment · Reviewer_oiRX · 2025-11-28
> > **Post-Rebuttal Comments**
> >
> > I appreciate the authors’ detailed rebuttal. The responses address my earlier concerns, and I am willing to raise my score.
> >
> > Considering the dataset with 22K trajectories and the proposed Gap-Aware Hybrid Reward, and in light of the growing use of large pretrained models (LLMs/VLMs) for decision-making in ObjectNav, I think the work is well aligned with current research trends and beneficial to the ObjectNav community.

---

### Author Response · Authors · 2025-12-02

**To the Area Chair and Reviewers,**

We thank the reviewers for their constructive feedback. During the rebuttal, we conducted significant additional experiments to address the raised concerns.

**1. Updated Reviewer Status**

Following our rebuttal and additional experiments, **3 out of 4 reviewers** have confirmed they are raising their scores:

* Reviewer iCB1: Stated "I would like to raise my score to  **strong accep**t " (Acknowledged new experiments on NavNuances and ProcTHOR).
* Reviewer oiRX: **Raising score**, satisfaction with the Stop Agent analysis and cost/scalability clarifications. They also highlighted the work's alignment with current MLLM trends.
* Reviewer Bopw: **Raising score**, citing the improved navigation understanding demonstrated by NavNuances.
* Reviewer 6mHY: (No response yet). We have addressed their concerns regarding the "Necessity of LVLMs" and "Reasoning Integration" in our detailed response.

**2. Key Improvements and Additional Experiments**

To validate "Decision Understanding" and "Generalization," we provided the following empirical evidence:

* Verified Decision Understanding:
  We evaluated CompassNav on the  NavNuances benchmark. Results (Table 1 in rebuttal Bopw) show a **~3x improvement** in spatial reasoning (Vertical Movement) over the base model and significantly outperform NavGPT-4V. This quantitatively proves our agent learns structural connectivity, not just path imitation.
* Proven Generalization on Unseen Domains:
  We extended evaluation to **AI2-THOR (ProcTHOR)** to test Sim-to-Sim transfer. CompassNav achieved a **+9.1% improvement in SPL** compared to baselines (Table 2 in rebuttal iCB1), proving robustness beyond the training domain (HM3D).
* System Reliability:
  We conducted a standalone confusion matrix analysis of the  Stop Agent. With a Precision of 88.3% and Recall of 69.4%, ensuring that our reported Success Rate (SR) is reliable and conservative.

**3. Addressing Methodology Concerns**

- E2E vs. Modular:
  We clarified to Reviewers 6mHY and oiRX that comparing CompassNav solely to engineered modular systems overlooks our contribution to the  **End-to-End LVLM paradigm** . In this setting, CompassNav achieves a **+36.8% performance leap** over the base model and outperforms proprietary models like GPT-4o-mini without external maps.

* Reasoning Integration (Response to 6mHY):
  We clarified that reasoning tokens in our GRPO framework are functionally integrated. The policy gradient ensures the model is incentivized to generate valid reasoning that directly leads to high-reward actions.

**4. Conclusion**
Given the new evidence for Decision Understanding and Generalization, along with the explicit support from three reviewers, we believe CompassNav offers a substantial contribution to Embodied Navigation. We remain available for further discussion.

---

### Meta-Review · Area_Chair_9Wo8 · 2026-01-10

**Summary:**

This paper proposes an SFT-RFT pipeline for VLN tasks, with major contributions being the Compass-Data-22k dataset and gap-aware hybrid reward mechanism. Reviewers raised a number of concerns, with important ones being demonstration of the claim that the model supports reasoning (raised by a number of reviewers), analysis of various elements such as the Stop Agent, and limitations in the experimental environments used. These are all relevant concerns that speak to validating the claims and contributions of the paper.

**Reviewer Concerns:**

Most of the significant concerns mentioned above were addressed with additional experimental results and analysis, especially demonstrating that the model does indeed contribute to increased reasoning (NavNuances results), analysis of the Stop Agent, etc. Overall the pipeline is somewhat standard as two-stage training and reasoning-based data generation is now common across a number of single and multi-modal domains, and performance in some cases are marginal over very hand-engineered modular architectures.

**Reviewer Scores:**

Three reviewers explicitly mentioned raising their scores, and based on the content of the rebuttal with respect to the concerns I do not have any reason to doubt these claims. Overall, while the paper has some limitations in novelty in terms of the overall framework, it does have significant contributions and interesting elements that the reviewers appreciated, and with the new experiments it has stronger evidence for the claims made regarding increased navigational reasoning. I do strongly encourage the authors to add these results in the final paper, and tone down some of the claims of "Path Imitation to Decision Understanding" as a new paradigm shift introduced by the paper, as there are now a number of works that have had a similar high-level motivation/design.

---

### Decision · Program_Chairs · 2026-01-26

Accept (Poster)